



# Dissolved organic matter characterization in soils and streams in a small coastal low-arctic catchment

Niek Jesse Speetjens[1], George Tanski[1,2,3], Victoria Martin[4], Julia Wagner[5], Andreas Richter[4], Gustaf Hugelius[5], Chris Boucher[1], Rachele Lodi[6], Christian Knoblauch[7], Boris P. Koch[8,9], Urban Wünsch[10], Hugues Lantuit[2] and Jorien E. Vonk[1]

[1]Vrije Universiteit Amsterdam (VUA), Department of Earth Sciences, Earth and Climate Cluster, Amsterdam, 1081 HV Amsterdam, The Netherlands
[2]Alfred Wegener Institute (AWI) Helmholtz Centre for Polar and Marine Research, Permafrost Research Unit, 14473 Potsdam, Germany
[3]Natural Resources Canada, Geological Survey of Canada–Atlantic, B2Y 4A2 Dartmouth, Canada
[4]University of Vienna (UniVie), Centre for Microbiology and Environmental Systems Science, Div. of Terrestrial Ecosystem Research, 1030 Wien, Austria
[5]Stockholm University (SU), Department of Physical Geography,106 91 Stockholm, Sweden
[6]Ca' Foscari University of Venice (Unive) and National Research Council, Institute of Polar Science (ISP-CNR), 30172 Mestre Venezia, Italy
[7]Universität Hamburg, Department of Earth Sciences, Institute of Soil Science, 20146 Hamburg, Germany
[8]Alfred Wegener Institute (AWI) Helmholtz Centre for Polar and Marine Research, Ecological Chemistry Research Unit, 27570 Bremerhaven, Germany
[9]University of Applied Sciences, An der Karlstadt 8, 27568 Bremerhaven, Germany
[10]Technical University of Denmark, National Institute of Aquatic Resources, Section for Oceans and Arctic, 2800 Kgs. Lyngby, Denmark

*Correspondence to*: n.j.speetjens@vu.nl, niek.j.speetjens@gmail.com and j.e.vonk@vu.nl

**Abstract.** Ongoing climate warming in the western Canadian Arctic is leading to thawing of permafrost soils and subsequent mobilization of its organic matter pool. Part of this mobilized terrestrial organic matter enters the aquatic system as dissolved organic matter (DOM) and is laterally transported from land to sea. Mobilized organic matter is an important source of nutrients for ecosystems as it is available for microbial breakdown, and thus a source of greenhouse gases. We are beginning to understand spatial controls on the release of DOM as well as the quantities and fate of this material in large arctic rivers. Yet, these processes remain systematically understudied in small, high-arctic watersheds, despite the fact that these watersheds experience the strongest warming rates in comparison.



Here, we sampled soil (active layer and permafrost) and water (porewater and stream water) from a small catchment along the Yukon coast, Canada, during the summer of 2018. We assessed the organic carbon (OC) quantity (using dissolved (DOC) and particulate OC (POC) concentrations and soil OC content), quality ($\delta^{13}$C-

DOC, optical properties, source-apportionment), and bioavailability (incubations, optical indices such as slope ratio (Sr) and humification index (HIX)) along with stream water properties (T, pH, EC, water isotopes). We classify and compare different landscape units and their soil horizons that differ in microtopography and hydrological connectivity, giving rise to differences in drainage capacity.

Our results show that porewater DOC concentrations and yield reflect drainage patterns and waterlogged conditions in the watershed. DOC yield (in mg DOC g soil OC$^{-1}$) generally increases with depth but shows a large variability near the transition zone (around the permafrost table). Active layer porewater DOC generally is more labile than permafrost DOC, due to various reasons (heterogeneity, presence of a paleo-active layer, and sampling strategies). Despite these differences, the very long transport times of porewater DOC indicate that substantial

processing occurs in soils prior to release into streams. Within the stream, DOC strongly dominates over POC, illustrated by DOC/POC ratios around 50, yet storm events decrease that ratio to around 5. Source-apportionment of stream DOC suggests a contribution of around 50 % from permafrost/deep-active layer OC, which contrasts to patterns observed in large arctic rivers (12 ±8 % Wild et al., 2019). Our 10-day monitoring period demonstrated temporal DOC patterns on multiple scales (i.e. diurnal patterns, storm-events, and longer-term trend) underlining

the need for high-resolution long-term monitoring. First estimates of Black Creek annual DOC (8.2 ±6.4 t DOC yr$^{-1}$) and POC (0.21 ±0.20 t yr$^{-1}$) export allowed us to make a rough up-scaling towards the entire Yukon Coastal Plain (34.51 ±2.7 kt DOC yr$^{-1}$ and 8.93 ±8.5 kt POC yr$^{-1}$). With raising arctic temperatures, increases in runoff, soil OM leaching, permafrost thawing and primary production are likely to increase the net lateral OC flux. Consequently, altered lateral fluxes may have strong impacts on the arctic aquatic ecosystems and arctic carbon

cycling.

## 1. Introduction

Global temperatures are rising and due to arctic amplification, surface air temperatures in high latitudes have increased by more than double compared to the global average (Meredith et al., 2019). Through numerous feedback loops, these climatological changes have significant impacts on both arctic and global biogeochemical

cycles, climate and ecosystems (AMAP, 2017). Perennially frozen ground (permafrost), underlying about a 18 %
of the exposed land surface area in the northern hemisphere (Zhang et al., 1999, 2008), experiences significant
warming and thaw (Biskaborn et al., 2019; Olefeldt et al., 2016). This is likely to have far-reaching consequences
on local arctic ecosystems and communities (Teufel & Sushama, 2019) as well as globally through the permafrost
carbon feedback on global climate (Koven et al., 2011; MacDougall et al., 2012; Schuur et al., 2015).


Permafrost soils store large amounts of organic matter (OM) (Hugelius et al., 2014). The relatively flat ice-wedge
polygon (IWP) tundra plains, which are thought to cover around 2,600,000 km$^2$ (Mackay, 1972), or roughly ~12.4
% of the northern permafrost domain (Obu et al., 2019), are particularly rich in OC due to enhanced storage and
preservation of soil organic carbon (SOC) caused by waterlogged anaerobic conditions (Zimov et al., 2006, Fritz
et al., 2016). Upon thaw, microbial and physical decomposition of permafrost OM releases greenhouse gases
(GHG's). Thawing permafrost OM is released into aquatic systems as dissolved (DOM) and particulate organic
matter (POM), which have varying pathways in the ecosystem. While parts of the thawed permafrost OM (i.e.
carbon and nutrients) can be degraded quickly or directly incorporated into living organisms, the more refractory
parts can be mineralized over longer timescales or sequestered via sedimentation or burial (McGuire et al., 2009;
Huissteden & Dolman, 2012; Vonk & Gustafsson, 2013; Knoblauch et al., 2013). Little is known about the
controls on lateral permafrost OM release and transport pathways from soils to aquatic systems (Fouché et al.,
2017; Vonk, Tank & Walvoord, 2019; Beel et al., 2020; Coch et al., 2020). Hence it is challenging to assess
landscape-scale flux variability and integrated estimates of lateral carbon fluxes and budgets are scarce in these
permafrost landscapes. It is therefore a priority to better understand lateral permafrost carbon dynamics and their
biogeochemical implications for Arctic ecosystems on a landscape level in order to include them into future
projection models.

Most studies investigating lateral OM and nutrient fluxes in the Arctic focus on the largest Arctic rivers (Ob,
Yenisey, Lena, Kolyma, Mackenzie, Yukon), whose watersheds drain about two-thirds (~67 %) of the total Pan-
Arctic watershed area (16.8 10$^6$ km$^2$) (e.g. Mann et al., 2012; Holmes et al., 2012; Wild et al., 2019). While the
watersheds of the six largest rivers drain vast areas of land, only 35 % of their catchments are underlain by
continuous permafrost. In contrast, the eight next largest ("Middle 8") and much smaller coastal catchments
draining to the Arctic Ocean (AO) are exclusively underlain by continuous permafrost ("Middle 8": 60 %,
"Remainder": 73 %) (Holmes et al., 2012). Smaller coastal watersheds experience a greater warming trend due to





their proximity to the coast where a rapidly-declining sea ice cover speeds up warming (Parmentier et al., 2013).
Despite this rapid warming and associated impacts on permafrost degradation (Olefeldt et al., 2016) and shifts in
hydrological and biogeochemical processes (Vonk et al., 2015), small watersheds remain understudied. Several
studies have however shown the relative importance of small watersheds in terms of discharge (e.g. Prowse &
Flegg, 2000; Lewis et al., 2010; Bring et al., 2016).


Permafrost DOM dynamics in small arctic watersheds are characterized by a high spatial and temporal variability
in terms of OM amount and composition and in terms of terrestrial imprint on stream DOM (e.g. Lewis et al.,
2012; Wauthy et al., 2018; Shatilla et al., 2019). Aquatic DOM fluxes are mainly driven by the underlying soil,
landform, climate and other environmental and chemical controls (Tank et al., 2020). Additionally, studies found

that land-ocean DOM fluxes in High Arctic watersheds are mainly controlled by thermal (active layer deepening)
and physical disturbances (e.g. retrogressive thaw slumps and thermoerosion) and increasingly impacted by
rainfall runoff instead of snowmelt (Beel et al., 2020; Lafrenière & Lamoureux, 2013). This is in good agreement
with studies in the western Canadian Arctic, where DOM fluxes increase (Coch et al., 2018) upon climate warming
and subsequent permafrost degradation. In contrast, there are also other studies suggesting a decrease in DOM

fluxes (e.g. Kawahigashi et al. 2004), resulting from increased mineral surface interaction as thaw depth increases
and disturbances become more frequent.

Here we focus on the Yukon coastal plain in the western Canadian Arctic (fig. 1), which is dominated by IWP
tundra with three IWP development stages: Low centered polygons (LCP), where ice wedges are large and intact,

high centered polygons (HCP), where ice wedges have already partially degraded, and flat polygons, which are
an intermediate stage in the transition from LCP to HCP (fig. 2). Wainwright et al. (2015) suggest that landscape-
scale classification by polygon type is useful in differentiating between lateral carbon flux magnitudes and
controls hence we follow this classification in differentiating soil characteristics. We targeted an unnamed creek
(referred to as Black Creek in this paper), draining a small IWP catchment (~ 4 km$^2$). Our objectives are to quantify

and characterize stream water DOM and POM (main stem, and tributaries) as well as porewater DOM (different
thermal layers, soil horizons, and polygon types) within this catchment using a combination of bulk isotopic and
optical techniques. Additionally, we performed incubation experiments to assess the potential lability of soil and
stream OM. We also investigated landscape heterogeneity in terms of polygons types within a catchment as it
represents a gradient from water-logged, anoxic, stagnant conditions (LCP) to drained and aerated soil conditions



(HCP), and the imprints of LCP and HCP on the stream water outflow characteristics and OM flux. These measurements present insights into the characteristics and dynamics of OM-release and controlling factors from predominant landscape types that characterize circumpolar small coastal watersheds.

In summary, the aim of this study is to better assess the role of small circum-Arctic watersheds to improve our

current understanding of land-ocean OM budgets. The specific objectives of this study are to (i) characterize the OM in the most dominant IWP types, (ii) investigate the degradation patterns of mobilized OM during transport from soil to stream (i.e. bioavailability), (iii) determine the quantity, character and origin of OM exported from the stream and ultimately (iv) to estimate the magnitude of annual OC exports from small streams on a landscape scale. With this study, we provide valuable data on so far understudied small watersheds and help to build a

baseline, which allows for better estimates of panarctic land-ocean OM fluxes.

## 2 Materials and methods

### 2.1 Study Area

Black Creek Watershed (BCW) is situated on the Yukon coastal plain in the western Canadian Arctic and drains into Ptarmigan Bay, which is a semi-open lagoon sheltered from the open Beaufort Sea (fig. 1). Black Creek is a

small coastal stream draining a polygonal tundra landscape underlain by continuous permafrost. The contributing watershed area is approximately 4 km$^2$, estimated using ArcticDEM, a publicly available 10-meter resolution digital elevation model (Morin et al., 2016; *accessed on May 28, 2020*), from which we obtained a watershed delineation using GRASS GIS. The Yukon coastal plain stretches ~300 km from the Mackenzie Delta in the East to the Alaskan border in the West. The Quaternary surficial geology is mainly characterized by lacustrine,

morainal, fluvial and colluvial deposits (Rampton, 1982). IWP tundra, moraine hills, wetlands, beaded streams and thermokarst lakes are the predominant landscape types. The land cover can be classified as low shrub tundra, subzone E (Walker et al., 2018) with occurrence of *Betula nana*, *Salix polaris*, mosses and lichens in HCP, while graminoids dominate in LCP terrain. The mean summer temperature (June, July and August 1991-2020) is 7.7 °C (± 4.6 °C) based on available data for three nearby stations (Herschel Island – Qikiqtaruk, Komakuk Beach and

Shingle Point). The mean annual temperatures at Shingle Point and Komakuk Beach are −9.9 °C and −11 °C respectively and precipitation means 254 and 161 mm (Environment Canada, https://climate.weather.gc.ca/climate_normals).



The region of interest is underlain by continuous permafrost and active layer depths average around 30-40 cm in
IWP terrain on nearby Herschel Island (Siewert et al., 2021). Ground ice volumetric content in the Yukon coastal
plain averages around 46 % but reaches as high as 74 % in some areas (Couture et al., 2018; Couture & Pollard,
2017). The warm season in the western Canadian Arctic, during which the stream network is active lasts
approximately four months (Dunton et al., 2006). On average the sea ice-break up in the southern Beaufort Sea
region starts around mid-May and freeze-up starts early October with prolonged open-water periods around the
Mackenzie delta area. Both, winter and summer sea ice extent and concentration gradually declined in recent
decades (Galley et al., 2016). The lengthening of the sea ice-free seasons leads to increased storm frequency and
intensity. In combination with higher surface temperatures in the region (Screen et al., 2012) these environmental
changes are expected to have a drastic impact on biogeochemical cycling and hydrological processes in the
western Canadian Arctic (Parmentier et al., 2017).

**2.2 Meteorological data**

During the sampling period (August 8$^{th}$ - 19$^{th}$ 2018), we collected on-site weather data at a 5-minute interval. Air
temperature was measured at 1.5 m above the ground (BTF11/002 TSic 506; ± 0.1 °C accuracy). Precipitation
was measured (Young Model 52203; 0.1 mm per tip; accuracy ± 2 %) at 0.5 m above the ground away from any
objects causing potential wind shadow. Wind speed was measured using Thies CLIMA (4.3519.00.173; ± 0.5 m
s-1 accuracy). Available weather data from outside the sampling period was downloaded from the Canadian
Government Environment and Natural Resources website (https://climate.weather.gc.ca) for the three nearby
stations mentioned in 2.1 (Station ID: 2100636, 2100682 and 2100950 respectively).

**2.3 Soil and water sampling and stream measurements**

Soil and water samples were collected between the 9$^{th}$ and 19$^{th}$ of August 2018. Soil samples were taken at 46
sites within the main polygon types in the watershed (HCP, LCP and flat polygon), which were classified based
on field observations. Both, the active layer (AL) and upper permafrost (PF) were sampled. Active layer samples
of known field volume were collected from the main soil horizon types (O, A, B, Bf/Cf) and classified according
to Schoeneberger et al. (2012). Samples with visible gley or cryoturbation were marked additionally. Permafrost
samples were collected at 10 cm intervals below the permafrost table up to a depth of 100 cm from the surface
(subject to practicality) using either a steel pipe and sledgehammer, SIPRE corer or Hilti drill hammer. All soil



samples (AL and PF; n = 153) were stored frozen at -18 °C in zip-lock bags until further processing in the lab, where porewater extraction took place. Stream water samples were taken every six hours at the catchment outlet using an ISCO 3700 automatic water sampler (Teledyne). In addition, manual samples were taken along the main channel and at three tributary streams flowing into the main stem using pre-rinsed 500 mL Nalgene bottles, which

were flushed with stream water three times prior to sampling. All water samples were filtered through pre-combusted and pre-weighed 47 mm glass fiber filters (GFF, Whatman, 47 mm diameter, 0.7 μm nominal pore size). Subsamples for DOC/$\delta^{13}$C-DOC analysis were acidified to pH < 2.0 using 36 % HCl (Suprapur) and stored cool (4 °C) and dark. Subsamples for chromophoric and fluorescent DOM (CDOM/fDOM) were stored frozen and dark at -18 °C. All filters with suspended material designated for total suspended solids (TSS), POC and POC-

$\delta^{13}$C analysis were stored frozen and dark. Basic hydro-chemical readings were taken at the stream outlet with an AP-5000 multiparameter probe (Aquaread), which was deployed at the catchment outlet from the 8th until the 19th of August. Measurements included relative water level (h [m]), water temperature (T [°C]), acidity (pH), electrical conductivity (EC [μS m$^{-1}$]), turbidity [NTU], dissolved oxygen (DO [% saturation]), redox potential [mV] and CDOM abundance [μg L$^{-1}$]. The CDOM sensor was calibrated using a quinine sulfate equivalent solution

(CDOM-CAL-600, Aquaread Ltd.) yet units are given in μg L$^{-1}$ as provided by the instrument. An empirical stage discharge equation was derived using flow measurements at different stages and fitting a quadratic function within the measured range to estimate discharge (Q [L s$^{-1}$]) as a function of the pressure head on the sensor (h [m]) of the stream (eq. 1).

$$Q = -21927h^2 + 7271.5h - 524.89 \qquad (1)$$

The measurement was based on the creeks flowing cross-sectional area together with flow velocity (measured at 2/3 of the water depth and 25 cm increments using a M1 mini current meter and Z6 counting device (SEBA Hydrometrie GmbH & Co. KG)) at the outflow at varying water levels.

**2.4 Porewater extraction**

Frozen soil samples were wet weighed and slowly thawed at 8 °C. Porewater was then extracted from active layer and permafrost samples (*n* = 142) using Rhizon samplers (mean pore size of 0.6 μm, Rhizosphere, Wageningen, The Netherlands) under cold and dark conditions in a cooler room (4 °C). Subsamples for CDOM/fDOM analyses





were taken from the extracted porewater and transferred into 15 mL falcon tubes and stored frozen and dark until

analysis. Subsamples for DOC/$\delta^{13}$C-DOC were acidified (pH< 2) and stored cool (at 4 °C) and dark in 40 mL

pre-combusted glass vials until analysis.

**2.5 DOC Concentration and Isotopes**

The DOC concentration and $\delta^{13}$C-DOC were measured with an Aurora 1030 DOC analyzer (OI Analytical, USA)

connected via a Conflow V to an Isotopic Ratio Mass spectrometer (IRMS, Delta V, Thermo Scientific, Germany)

at the University of Hamburg (Germany) (stream water incubation samples), the Alfred Wegener institute for

Polar Research (Bremerhaven, Germany) (bulk porewater samples) following Hölemann et al. (2021) and at North

Carolina State University (Raleigh, USA) (stream water bulk samples and porewater incubation samples). DOC

concentrations were used to quantify the total amount of OC in a dissolved state within the watershed systems

and $\delta^{13}$C-DOC to derive the origin and relative degradation state of OM. DOC concentrations from porewater

were used to calculate yields in mg DOC g$^{-1}$ soil dry weight and mg DOC g$^{-1}$ soil organic carbon (SOC) using Eq.

2 an Eq. 3.

$$Y_{soil} = \frac{[DOC] \times m_w}{m_s \rho_w} \tag{2}$$

$$Y_C = \frac{[DOC] \times m_w}{m_C \rho_w} \tag{3}$$


Where $m_w$ is the mass in grams of water present in the sample, measured as the difference of the bulk wet soil

sample weight minus the dry weight and assuming the density of water at 4 °C as $\rho_w = 1 \cdot 10^3$ g L$^{-1}$. $m_s$ is the soil

bulk dry weight and $m_C$ is the total mass of SOC both in grams. SOC was measured as the fraction of the dry bulk

weight lost on combustion at 550 °C multiplied by a conservative factor of 0.45 to convert from bulk soil organic

matter (SOM) to SOC, in accordance with Jensen et al. (2018).

**2.6 POC concentration and Isotopes**

Carbonates were removed from freeze-dried filters using acid-treatment. For this, filters were subsampled (16

filter punches of 4 mm cross-section) into silver capsules, moisturized with 25 µL of distilled water, acidified

with 25 µL of 1 M HCl and left for 30 minutes at room temperature. Then, 50 µL of HCl was added and samples



were dried in an oven for 3 hours at 60 ºC. After that, silver capsules were folded and analyzed for % OC, % N, and $\delta^{13}$C (‰ VPDB) at the Vrije Universiteit Stable Isotope Laboratory (Amsterdam, The Netherlands).

### 2.7 Stable water isotopes of stream and porewater samples

Deuterium and oxygen isotopes ($\delta$D and $\delta^{18}$O) were measured on water subsamples with a Continuous Flow Deltaplus X IRMS coupled to a Flash Elemental Analyzer and High Temperature Conversion Elemental Analyzer

(TC/EA) at Vrije Universiteit Amsterdam and are given as ‰ difference from Vienna Standard Mean Ocean Water (VSMOW). Deuterium excess was used to allocate the precipitation source (e.g. Fritz et al., 2016) in the same region (d-excess = $\delta$D−8$\delta^{18}$O; Fritz et al., 2016, Dansgaard, 1964). Data was compared with water isotopic values from two local meteoric water lines (LWML) in Inuvik (200 km south-east) ($\delta$D= 7.3 * $\delta^{18}$O -3.5) (Fritz et al., 2016) and Utqiagvik (former Barrow), Alaska (600 km north-west) ($\delta$D = 7.5 * $\delta^{18}$O − 1.1) (Throckmorton

et al., 2016).

### 2.8 DOM optical properties

The chromophoric fraction of DOM was used to characterize DOM and identify the source. The CDOM and fDOM (fluorescent DOM fraction) were used as indicators of DOM quality such as degradation status, molecular size, and DOM source (Stedmon & Nelson, 2015). We used a range of absorbance and fluorescence indices for

characterization of DOM, summarized in Table 1. Fluorescence data was processed using drEEM toolbox (Murphy et al., 2013) in MATLAB (R2017b). CDOM and fDOM were measured on a Horiba Aqualog fluorescence spectrophotometer at the Technical University of Denmark (DTU, Copenhagen).

### 2.9 DOM lability

In this study we chose four indicators to estimate degradation state and to infer lability:


1. Slope ratio (Sr), calculated as the ration between the slope of the absorbance between 275–290 nm and 350-400 nm and absorbance ratio ($a_{254}$:$a_{365}$) have been recognized as indicators of DOM molecular weight (MW) (Helms et al., 2008). The assumption is that lower MW organic molecules will generally be more bioavailable;

2. The specific UV absorbance at 254 nm divided by the DOC concentration has been identified as a proxy for DOM aromaticity (Weishaar et al., 2003). Fouché et al. (2020) show that high Sr (i.e. low



molecular weight) and low SUVA$_{254}$ (i.e. low aromaticity) together are indicative of higher lability of permafrost DOM;

3. The degradation status, which can be inferred from the Humification index (HIX), calculated as the

area under the emission spectra 435–480 nm divided by the peak area 300–345 nm + 435–480 nm, at excitation wavelength 254 nm (Ohno, 2002). DOM degradation in soils is the processing of labile fresh organic products (e.g. sugars) to more chemically complex and less bioavailable forms (Balser, 2005). Hence, more degraded DOM will generally be less labile.

4. Freshness index (FRESH) calculated from emission intensity at 380 nm divided by the maximum

emission intensity between 420 nm and 435 nm at excitation 310 nm (Parlanti et al., 2000; Wilson & Xenopoulos, 2009) and fluorescence index (FI) the ratio of emission intensity at wavelength 470 nm and 520 nm, at excitation wavelength 370 nm which indicate what proportion of DOM is likely to be fresh and microbially produced (McKnight et al., 2001; Cory et al., 2010) of microbial origin. The assumption is that small, fresh, microbial leachates (i.e. high FI and FRESH) correlate with higher DOM lability.


Additionally, we performed incubations with stream and porewater samples to estimate the degradation potential of DOC by comparing DOC concentrations and δ$^{13}$C-DOC values before and after incubations.

### 2.9.1 Stream water incubation

For three tributary streams three aliquots of 60 mL water were incubated (in the field) for 14 days at ambient air

temperature of ~4 °C under dark and oxygenated conditions in 120 mL amber glass vials. These incubations were repeated at three different instances during the field campaign. Samples were turned regularly to prevent flocculation and mimic mixing in the stream. Incubations were stopped after two (T=2), seven (T=7) and 14 days (T=14). At each time step, samples were filtered using pre-combusted glass fiber filters (GFF, nominal pore size 0.7 µm). The filtrate was split into subsamples for DOC/δ$^{13}$C-DOC analysis (acidified to pH <2.0 with 36 % HCl

(Suprapur) and stored dark at 4 °C), and subsamples for CDOM/fDOM (stored dark and frozen at -18 °C).

### 2.9.2 Porewater incubation

We incubated porewater extracts from six upper active layer samples (Oi-horizon) and nine upper permafrost samples at the Vrije Universiteit in Amsterdam to check for differences in OM degradation potentials between active layer and permafrost. Incubations were conducted under laboratory conditions following procedures



adapted from Vonk et al. (2013) and Spencer et al. (2015). Rhizon-filtered samples (median pore size 0.6 µm) were transferred to pre-combusted 40 mL amber glass vials, holding in total 30 mL of sample. An inoculum was added and prepared from a soil slurry consisting of a total of 12 g (2 g of each) of the sampled Oi-horizons mixed with 240 mL of autoclaved tap water. The slurry was filtered through a glass fibre filter (Whatman, 1.2 µm nominal pore size) and 1 mL was added to each incubation sample. Samples were then placed on a shaker table

for incubations at 8 °C under dark and oxygenated conditions. The incubations were stopped after T=2, 7, 14 and 21 days by acidification to pH < 2.0 (using 36 % HCl, Suprapur). Samples were stored cool (4 °C) until further analysis for DOC concentration and $\delta^{13}$C-DOC.

### 2.10 Endmember mixing model

Along with isotope tracers such as $\delta^{13}$C-DOC, $\Delta^{14}$C-DOC, $\delta$D and $\delta^{18}$O (Vonk et al., 2012; Grotheer et al., 2020),

absorbance and fluorescence properties have been successfully used for source apportionment approaches and to characterize DOM and trace sources (Lee et al., 2020). We used Endmember Mixing Model Analysis (EMMA) to estimate the contribution of three potential sources (permafrost, active layer, and in-stream primary production) to the Black Creek stream using $\delta^{13}$C-DOC, Sr and $a_{254}/a_{365}$ as tracers. These parameters are considered semi-conservative and most distinctive in separating endmembers. We used a Bayesian mass-balance source

apportionment model with Metropolis-Hastings Markov Chain Monte Carlo sampling following Bosch et al. (2015) in MATLAB R2017b. To compute source contributions, we ran the model with a time series of measured $\delta^{13}$C-DOC, Sr and $a_{254}/a_{365}$ at the catchment outlet over time. Although sample size is limited due to practical limitations, we chose to use $\delta^{13}$C-DOC of porewater instead of $\delta^{13}$C-SOC of soils to avoid fractionation effects from soil-to-water leaching (e.g. Kaiser et al., 2001; Boström et al., 2007) impacting the source-apportionment.

For active layer, we used a $\delta^{13}$C-DOC value of -26.4 ±1.07 ‰ based on the porewater DOC-$\delta^{13}$C measurements in the catchment (n = 3). Active layer values for Sr and $a_{254}/a_{365}$ were also based on porewater samples from the catchment and were 0.71 ±0.08 (n = 45) and 4.55 ±0.8 (n = 45), respectively. For permafrost we used a $\delta^{13}$C-DOC value of -24.15 ±1.03 ‰ (n = 6), and Sr and $a_{254}/a_{365}$ of 0.85 ±0.08 (n = 44) and 5.81 ±1.2 (n = 45) respectively, based on $\delta^{13}$C-DOC and Sr and a254/a365 of permafrost porewater. Finally, the primary production source was

set to -28.48 ±1.0 ‰ (n = 9) based on $\delta^{13}$C-DOC values of the tributaries where we observed primary production (algal mats). The standard deviation of the analyzed endmember samples was 0.237 ‰. However, we acknowledge that the tributary water $\delta^{13}$C-DOC signal probably consists of a mixture of terrestrial and primary
production source leachates. Moreover, it is likely that fractionation takes place during leaching from OM sources, including primary production sources. Hence, we expect a pure primary production signal would be more depleted

than that found in the tributary water and we have therefore increased the standard deviation to ±1 ‰, similar to the other $\delta^{13}$C-DOC sources, to account for uncertainty. The Sr and $a_{254}/a_{365}$ of the primary production endmember were set at 0.78 ±0.020 (n = 9) and 5.40 ±0.23 (n = 9) respectively. To estimate mixing with terrestrial DOC we also ran the simulation with $\delta^{13}$C-POC (32.68 ±2.00 ‰, n = 9) instead of $\delta^{13}$C-DOC by means of sensitivity analysis. We acknowledge that using the $\delta^{13}$C-POC to trace DOM is not incorporating potential fractionation

efforts through leaching and that the most realistic primary production $\delta^{13}$C value probably lies between the $\delta^{13}$C-DOC and $\delta^{13}$C-POC values used here.

We performed a sensitivity analysis by increasing and decreasing (±5 %) permafrost endmember tracer means and standard deviations separately and comparing the effect (percentage change) on relative contribution of each

source in the mixing model. The sensitivity analysis focused on permafrost endmember values since we are primarily interested in the relative contribution of thawing permafrost. Additionally, it is assumed that changing other endmember tracer mean and standard deviation by the same order of magnitude will result in same order of magnitude changes in relative source contribution and testing with only permafrost is representative of all endmembers.

**2.11 Statistical analyses**

All statistical analyses were performed in Python 3 programming environment (Van Rossum & Drake, 2009) using pandas (McKinney, 2010), scipy (Jones et al., 2015) and statsmodels (Seabold & Perktold, 2010) packages. Significance statistics were calculated with a two-sided T-test. Linear regression results were obtained using linear least-squares regression for two sets of measurements as described in the scipy manual.

**3 Results**

**3.1 Meteorology and hydro-geochemistry**

Weather conditions during the field campaign were variable with air temperatures ranging between -0.8 °C and 12.7 °C (*mean:* 4.2 ±2.6 °C) (fig. S1). The predominant wind direction was northwest with mean wind speeds of 4.78 ±2.92 m s⁻¹ and gusts up to 15.1 m s⁻¹ (7 Bft). Precipitation was generally low with notable rainfall recorded





on August 13 (1.3 mm) and between August 16 and 19 (7.7 mm cumulative). Total precipitation during the
       monitoring period was 9.8 mm (table S1). Mean electrical conductivity (EC) at the catchment outlet was 954 μS
       cm$^{-1}$, pH ranged between 6.5 and 7.8 (*mean* = 6.9 ± 0.19) and water temperature ranged between 2.9 °C and 12.3
       °C (*mean* = 6.6 ± 1.7 °C). During a storm event on the 16[th] and 17[th] of August water levels at the outlet monitoring
       station peaked together with EC (19134 μS cm$^{-1}$). Discharge in the creek main channel showed a decreasing trend

(*min* = 9, *max* = 140, *mean* = 43 ± 27 L s$^{-1}$) during the measurement period. Discharge during the storm event of
       the 16[th] and 17[th] was disregarded due to uncertainty regarding tidal influence, which is indicated by the elevated
       EC values. Water isotope values within the main channel ranged between -130.3 ‰ and -127.1 ‰ (δD) and -16.35
       ‰ and -16.25 ‰ (δ$^{18}$O) with mean values of -128.0 ± 1.2 ‰ and 16.32 ± 0.03 ‰ respectively. In tributary streams
       stable water isotope values covered a wider range, between -132.4 ‰ and -94.38 ‰ (δD) and -17.11‰ and -

11.94‰ (δ$^{18}$O) with mean values of -122.9 ± 7.5‰ and 15.71 ± 0.98‰, respectively. Stable water isotope signals
       were grouped by main source (i.e. permafrost and active layer porewater, tributaries, main channel) (table 2). The
       correlation of δD and δ$^{18}$O for the samples compared to the local meteoric water line (LMWL) in Inuvik shows
       that permafrost samples group in roughly three lines that are distinguished by their d-excess (fig. S2). The majority
       of the permafrost samples plot further from the LMWL at Inuvik than the modern samples (streams and active

layer), which in turn also are deviating from the Inuvik LMWL.

### 3.2 OC concentrations and stable isotopes

### 3.2.1 Concentrations and δ$^{13}$C of DOC and POC in streams

The DOC concentrations in the main channel upstream and at the outlet were on average 13.3 ± 2.04 mg L$^{-1}$ and
16.0 ± 3.25 mg L$^{-1}$, respectively (fig. 3, table S2). At the outlet, values below 5 mg L$^{-1}$ were measured during the

storm event of August 16[th] and 17[th]. DOC concentrations correlated with measured CDOM (Aquaread's
       Aquaprobe AP-5000) concentrations (133.7 ± 29.02 μg L$^{-1}$; fig. 3, table S2), and showed a gently declining trend
       over the monitoring period. The average DOC concentrations measured in tributaries were significantly higher ($p$
       < 0.05) (22.74 ± 8.66 mg L$^{-1}$) than in the main stem. POC concentrations and δ$^{13}$C-POC values in the main channel
       (0.41 ± 0.2 mg L$^{-1}$, -29.31 ± 0.829 ‰) were significantly different ($p$ < 0.05) to the values in the tributary streams

(1.16 ± 0.604 mg L$^{-1}$, -32.68 ± 2.042 ‰, Table 3). The POC concentrations increased during the storm event,
       opposite to DOC concentrations which declined. Similarly, during this event the δ$^{13}$C-POC signal became less
       depleted whereas the δ$^{13}$C-DOC signal became slightly more depleted. The tributaries had the most depleted δ$^{13}$C-





DOC signal (-28.476 ±0.2368 ‰) of all samples. In the tributaries we observed algal production. The $\delta^{13}$C-POC signal of the filters taken from this tributary were the most depleted found during this study (-35.25 ±1.00 ‰, n=3).

### 3.2.2 Concentrations, yields, and $\delta^{13}$C of DOC and SOC in soils

SOC contents (% of dry weight) of the sampled soils were high but differed strongly (fig. S3, table S3) between LCP active layer (26 ±13.2 %, $n$ = 16) and LCP permafrost (17 ±10.7 %, $n$ = 21) while differences for HCP active layer (23 ±12.8 %, $n$ = 38) and HCP permafrost (18 ±6.2 %, $n$ = 44) as well as flat active layer (17 ±13.5 %, $n$ = 12) and flat terrain permafrost (15 ±7.7 %, $n$ = 11) were less pronounced. Differences between permafrost and active layer were significant between LCP and HCP classes ($p < 0.05$) but not for flat type polygons. Although the differences between the three landscape classes within each thermal layer were large, they were not significant ($p > 0.05$).

The DOC concentration in porewater extracts showed a great variability (142.3 ±83.62 mg L$^{-1}$) but were significantly ($p < 0.05$) higher in permafrost (181.3 ±82.86 mg L$^{-1}$) compared to the active layer extracts (88.96 ±47.55 mg L$^{-1}$). Among permafrost samples, a significant concentration difference was found between HCP (171.19 ±87.8 mg L$^{-1}$) and LCP (150.61 ±63.1 mg L$^{-1}$) as well as HCP and flat (146.26 ±89.5 mg L$^{-1}$) type polygons ($p < 0.05$) with HCP having the highest concentrations (fig. 4a). Active layer DOC concentrations were not significantly different between polygon types.

DOC yields (Fig. 4c,d) were highly variable both in permafrost (0.27 ± 0.27 mg g$^{-1}$ soil, 2.10 ± 2.65 mg g$^{-1}$ C) and active layer soils (0.22 ± 0.31 mg g$^{-1}$ soil, 2.97 ± 6.69 mg g$^{-1}$ C), but were not significantly different from each other. While permafrost DOC yield is higher at lower SOC content, active layer DOC yield is highest with higher SOC (fig. 4c,d). Results also show that gleyed soils have slightly higher DOC yield (2.54 ±1.82 mg g$^{-1}$ C) compared to other active layer samples (2.31 ±1.85 mg g$^{-1}$ C). DOC yields above 7.10 mg DOC g$^{-1}$ C were considered as outliers since they were not within the 95-percentile range and therefore removed from the yield analyses. The DOC yields of permafrost samples are generally increasing with depth. Although DOC concentrations were significantly different ($p < 0.05$) between permafrost and active layer samples as well as between HCP permafrost and LCP permafrost, DOC yields were similar among the different classes (fig. S4).





This is similar for DOC concentrations which were not significantly different ($p > 0.05$) between polygon types in the active layer.

The $\delta^{13}$C-DOC values varied significantly ($p < 0.05$) between most sample sources. Values were highest and most
variable in HCP permafrost (-23.68 ±1.2 ‰) followed by LCP permafrost (-25.05 ±0.5 ‰). LCP permafrost $\delta^{13}$C-DOC was not significantly different ($p = 0.0519$) from the $\delta^{13}$C-DOC signal in the main channel (-25.40 ±0.4 ‰). In contrast to that, the $\delta^{13}$C-DOC signature in HCP active layer was more depleted and had a wider range (-26.38 ±1.1 ‰). The SOC-$\delta^{13}$C values in the catchment were generally more depleted than porewater $\delta^{13}$C-DOC (table 3); for permafrost and active layer in HCP (PF: -26.91 ±1.0624 ‰, AL: -27.41 ±0.9 ‰), LCP (PF: -27.01 ±1.0
‰, AL: -28.27 ±1.0 ‰) and flat (PF: -27.73 ±0.9 ‰, AL: -27.93 ±0.9 ‰). The difference between porewater $\delta^{13}$C -DOC and soil $\delta^{13}$C-SOC is larger for permafrost (LCP: 2.25 ‰, HCP: 3.7 ‰) compared to active layer (HCP: 1.21 ‰, LCP: n/a (tables S3 and S4).

### 3.3 Degradability of DOC

Incubation experiments show a high variability in DOC loss as well as $\delta^{13}$C-shifts for active layer and permafrost
(fig. 5). The degradability of DOC strongly differed between samples. The mean loss of DOC [%] from incubations with water from the three different tributary locations (A, B and C) was 4.30 ±5.3 % after 7 days of incubation. During this period, mean isotopic values of $\delta^{13}$C-DOC did not change significantly ($p > 0.05$) (-28.48 ±0.2 ‰ at T = 0 days and -28.52 ±0.3 ‰ at T = 7 days). However, looking at individual samples, $\delta^{13}$C-DOC enrichment was observed in several cases (table S3, fig. S5). Location A and B showed comparable patterns with
depletion of $\delta^{13}$C in the first 7 days and repletion between day 7 to 14, location C showed almost a linear trend toward a less depleted $\delta^{13}$C-DOC signal over time. The incubation of porewater showed that DOC losses are in the same range as the tributaries and that there are no significant differences between HCP permafrost (5.0 ±6 % DOC loss, n = 3, T = 7 days) and LCP permafrost (7.0 ±1 % DOC loss, n = 2, T = 7 days) ($p > 0.05$). In contrast to that, active layer DOC from concentration the Oi-horizon, although much lower in, showed significantly higher
relative DOC losses (17.3 ±16 %, n = 3, T = 7 days). The $\delta^{13}$C-DOC signals are significantly different between active layer, LCP permafrost and HCP permafrost at the begin of the incubation (T=0) and, on average, become





slightly more enriched over time. Yet, there is no significant change in $\delta^{13}$C-DOC between T=0 and T=21 (table 4, fig. S6).

### 3.4 Optical properties of DOM in the catchment

Values of SUVA$_{254}$ at the catchment outlet varied between ~1.71 and ~11.73 L mg$^{-1}$ C m$^{-1}$ and averaged around ~3.68 ±2.3 L mg$^{-1}$ C m$^{-1}$ (n = 27). The highest values of SUVA$_{254}$ correspond with low concentrations of DOC during the storm event (August 16$^{th}$ and 17$^{th}$). With exclusion of these storm event extremes, mean SUVA$_{254}$ values were slightly lower (~3.07 ±0.5 L mg$^{-1}$ C m$^{-1}$) and not significantly ($p > 0.05$) different from mean SUVA$_{254}$ values in tributaries (~3.86 ±1.8 L mg$^{-1}$ C m$^{-1}$). Permafrost porewater showed significantly lower SUVA$_{254}$ (~1.00

±0.6 L mg$^{-1}$ C m$^{-1}$) ($p < 0.05$) than active layer porewater (~2.13 ±0.9 L mg$^{-1}$ C m$^{-1}$). Highest mean active layer porewater SUVA$_{254}$ is found in HCP (~2.4 ±0.9 L mg$^{-1}$ C m$^{-1}$) followed by flat (~2.14 ±0.9 L mg$^{-1}$ C m$^{-1}$) and LCP (~1.55 ±1.1 L mg$^{-1}$ C m$^{-1}$). For permafrost the order was different, HCP showing the lowest values for SUVA$_{254}$ (~0.91 ±0.7 L mg$^{-1}$ C m$^{-1}$) followed by LCP (~1.06 ±0.6 L mg$^{-1}$ C m$^{-1}$) and flat polygon type showing highest SUVA$_{254}$ (~1.23 ±0.6 L mg$^{-1}$ C m$^{-1}$) in permafrost. Slope ratio (Sr), which negatively correlates with MW of DOM, was negatively correlated with SUVA$_{254}$ in our porewater samples while Sr shows a weak positive

correlation with SUVA$_{254}$ (i.e. lower MW molecules were less aromatic). In stream water samples, however, Sr was positively correlated with SUVA$_{254}$ (i.e. lower MW molecules were more aromatic). For fluorescence index and biological index (FI, BIX), which are used to indicate terrestrial sources of DOM (i.e. soil organic matter and plant litter/low FI and BIX) and relatively fresh, more microbially-derived DOM (i.e. leachates and products of

algae and bacteria/high FI and BIX) (Fouché et al., 2017) the same holds: in porewater samples the aromaticity was highest in samples with a terrestrial and less fresh DOM signal, while in streams the aromaticity was highest in samples with relatively fresh DOM signal.

Overall, our results show a predominantly terrestrial signature in streams which is slightly but significantly ($p <$

0.05) higher in tributaries (FI: ~1.47, BIX: ~0.50, Sr: ~0.80) compared to the outlet (FI: ~1.49, BIX: ~0.55, Sr: ~0.85). This signal is comparable to that of porewater in the active layer (FI: ~1.46, BIX: ~0.48, Sr: ~0.81). In contrast, the permafrost porewater averages around a more fresh, microbial and lower MW signature (FI: ~1.65, BIX: ~0.62, Sr: ~0.93). Permafrost and active layer are significantly different with respect to their SUVA$_{254}$, FI, BIX, and Sr values ($p < 0.05$). However, when looking closer at the distribution of their values within the soil

profile, we observe linear trends with depth respective to the permafrost table rather than clustering linked





explicitly to each thermal layer (fig. S7). With respect to polygon types, results show that HCP-active layer spectral indices are significantly different ($p < 0.05$) from the other polygon classes except for HIX and that differences between active layer and permafrost are most pronounced in HCP (table S6, fig. 6). The humification index (HIX) peaks in HCP active layer (~0.86), while lowest HIX was found in HCP permafrost (~0.75) (table

5). A more detailed peak in HIX around the permafrost table suggests the prevalence of more degraded SOM (fig. S7). With deeper permafrost sampling depth, a decreasing HIX value is observed. Although distinctly different, values of HIX are neither significantly different between active layer and permafrost nor between landscape classes. Values of HIX were highest in streams (~0.95) and the active layer (~0.85).

### 3.5 Endmember-based source apportionment of DOM

With our initial source apportionment setup, we modelled source contribution with mean isotopic (fig. S8) and spectral index values at the outlet ($\delta^{13}$C-DOC: -25.40 ‰, $a_{254}/a_{365}$: 5.55 and Sr: 0.86) for three endmembers (Permafrost OM, Active layer OM and primary production OM). Model results given these inputs indicate a relative contribution of these sources of ~48 ±20 %, ~30 ±21 % and ~22 ±15 % respectively (table 6, fig. S9, S10). We used the maximum (-26.08 ‰) and minimum (-24.73 ‰) $\delta^{13}$C-DOC values and corresponding Sr and

$a_{254}/a_{365}$ at the outlet to assess variation of source contribution over time. Correspondingly, we find that permafrost, active layer and primary production OM contributions vary with between 31-67 %, 20-38 % and 14-31 % respectively. When calculating the source apportionment considering HCP permafrost and LCP permafrost as different DOM sources (HCP permafrost $\delta^{13}$C-DOC: -23.68 ±1.2 ‰, Sr: 0.94 ±0.1, $a_{254}/a_{365}$: 6.08 ±1.1 and LCP permafrost $\delta^{13}$C-DOC: -25.05 ±0.5 ‰, Sr: 0.92 ±0.01, $a_{254}/a_{365}$: 5.67 ±1.4), the mean relative contribution of

permafrost sources increases from 48 % to 58 % at the catchment outlet. When using $\delta^{13}$C-POC of the primary production sources instead of $\delta^{13}$C-DOC, this leads to a decrease in primary production contribution from 21 % to 11 %, mostly resulting in an increase of permafrost contribution from 48 % to 55 % and an increase of active layer contribution from 31 % to 34 % (table 6 and 7). A summary of the time series and computed source contributions using $\delta^{13}$C-DOC of primary production can be found in table S9.


From sensitivity analysis (i.e. changing input parameters with fixed relative amounts) we observed that modelled source contributions respond strongest to shifts in $\delta^{13}$C-DOC. When decreasing the permafrost mean $\delta^{13}$C-DOC from -24.15 ‰ to -25.36 ‰, this resulted in a shift from 49 % to 52 % of relative contribution while active layer





contribution remains constant and primary production decreases from 21 % to 17 %. Inversely, when using a

higher permafrost mean $\delta^{13}$C-DOC (-24.15 ‰ to -22.94 ‰) this results in a decrease of its contribution from 49 % to 40 % while primary production increased from 21 % to 27 % (table S7 and S8) and active layer increased from 31 % to 33 %. Changing standard deviations of permafrost endmember values by ± 5 % leads to changes in contribution ranging from - 3 % for permafrost to + 5 % for primary production.

### 4 Discussion

The aim of this study is to better assess the role of small circum-Arctic watersheds to improve land-ocean OM budgets. The specific objectives of this study are to (i) characterize the OM in the most dominant IWP types (4.1), (ii) investigate the degradation patterns of mobilized OM during transport from soil to stream (4.2), (iii) determine the quantity, character and origin of OM exported from the stream (4.3) and ultimately (iv) to estimate an annual OC export from small streams on a landscape scale (4.4).

**4.1 Differences of OM pools in HCP and LCP**

Our data show that there are significant differences in terms of their OM pools between the two thermal layers (i.e. active layer and permafrost) and between the main landscape features that define the terrain (i.e. HCP and LCP). The main differences between HCP and LCP are the micro-topography and hydrological pathways, which may influence OC characteristics. The wetter soils in LCP have higher thermal conductance, hence summer active

layer depths in the center of the polygon often reach deeper than in HCP (Liljedahl et al., 2016, Walvoord & Kurylyk, 2016, Wales et al., 2020). LCP's elevated ice wedge rims generally promote waterlogged conditions in the polygon center while in HCPs, the degraded ice wedges form connected drainage channels resulting in well drained polygon centers (Liljedahl et al., 2012).

Differences in drainage patterns are reflected in our observed DOC concentrations and yields. Mean SOC contents are higher in the active layer (LCP: 26 ±13.2 %, HCP: 23 ±12.8 %) compared to the permafrost (LCP: 17 ±10.7 %, HCP: 18 ±6.2 %). This contrasts with the DOC concentrations, which are higher in permafrost (HCP: 171.20 ±87.8 mg L$^{-1}$, LCP: 150.62 ±63.1 mg L$^{-1}$) compared to active layer (HCP: 95.20 ±53.7 mg L$^{-1}$, LCP: 92.98 ±53.2 mg L$^{-1}$). The high mean DOC concentrations in permafrost porewater (162.44 ±82.0 mg L$^{-1}$) compared to those

of active layer (97.14 ±52.54 mg L$^{-1}$) indicate that the permafrost DOC pool is still freeze-locked (i.e. immobile)





whereas the active layer has been flushed more regularly. Differences in DOC concentration between active layer and permafrost were highest in HCP and lower in the LCP polygon type category. This could indicate that in the active layer of HCPs, DOM was already subject to more degradation and flushing with runoff. This is also suggested in a simulation study by Liljedahl et al. (2012) showing that in LCP terrain 46 % of the snow water equivalent was flushed as runoff while this was 73 % in HCP terrain.

Overall our data show a trend of increasing DOC yield (mg DOC g$^{-1}$ SOC) with increasing depth in the soil. The depth below the permafrost table showed to be a good indicator of DOC concentration as well ($[DOC] = 2.028 \cdot \Delta z_p + 134.60$, $R^2 = 0.71$, $p = 8.6e^{-37}$). The SOC-content showed a negative correlation with depth (fig. 7), dipping around the permafrost table. This 'dipping' effect could be ascribed to the increasing likelihood of waterlogging conditions to occur near the permafrost table of HCP soils (Harden et al., 2012). Signs of stagnant water were observed in several gleyed (typical brown/orange to grey/blueish colour patterning caused by waterlogged anoxic conditions) soil profiles. We also noted that in gleyed soil samples, comprising mostly mineral soils (B-horizons), there often was a low SOC content relative to the DOC concentration (i.e. a high DOC yield in mg-DOC/g C). This suggests that these soils, due to their waterlogged conditions, are either efficiently leached out and/or DOC is flushed in from overlaying O-horizons and accumulates. The $\delta^{13}$C signature of DOC shows a more degraded (i.e. more enriched) signal than the bulk SOC-$\delta^{13}$C, indicating that the degraded fraction of SOM is preferably leached and/or sorption processes are affecting $\delta^{13}$C-DOC signatures. Moreover, we find that values of permafrost $\delta^{13}$C-DOC are more enriched than those of active layer, indicating a more processed OM pool, or a stronger DOC leaching effect on $^{13}$C in mineral permafrost horizons than organic surface horizons.

### 4.2 Mobilization and degradation dynamics of OM from soils to streams

Our incubation experiments suggest that active layer DOC shows a higher lability (17 % DOC loss in 7 days) than DOC found in permafrost (HCP: 5 % DOC loss, LCP 7 % DOC loss in 7 days). In a meta-analysis, Vonk et al., (2015) calculated average bio-labile DOC (BDOC) content in permafrost of ~16 % after 28 days of aerobic incubation (T = 15-25 °C, n = 205) and even higher BDOC content when looking at continuous permafrost zone leachates only, based on several studies. Although our incubations were done at lower temperatures (4 °C), we observed that degradation rates stagnated after ~14 days, hinting that most labile BDOC would have been processed by then. Selvam et al. (2017), who incubated permafrost peat, showed much lower lability (~3 % after 7 days), which confirms that the bioavailability of DOM in permafrost is variable. It is worth noting that the





maximum depth at which permafrost was sampled in the study by Selvam et al. (2017) was only 5 cm below the permafrost table, comparable to our study. Vonk et al., (2015) could not include depth in their assessment of permafrost DOM lability due to limitations in the data. However, they do acknowledge the linkage between depth and DOM character and showed highest lability in deep Yedoma layers. In the following paragraphs we present

several explanations for the bioavailability of our samples.

### 4.2.1 Low DOC concentration with high bioavailability in the upper active layer

Our results show high variability in OM composition and quantity within the active layer. This is reflected in the DOC concentrations as well as the variability of its spectral properties. Oi-horizons in HCP showed relatively high SOC content and low DOC concentrations. We interpret this as a high abundance of fresh plant material,

which leaches less than more degraded plant material. Fresh DOM consists primarily of bioavailable sugars and products of microbial activity such as amino acids (Balser, 2005) which could explain why our Oi-DOM incubations showed relatively high BDOC loss compared to permafrost samples.

### 4.2.2 Inversed degradation status with depth in permafrost

More degraded DOM contains more humic-like substances (Ohno, 2002). Hence, as is expected in soil profiles,

an increasing degradation status with depth is reflected in our results (fig. S7). We see a (weak) increasing trend in HIX (i.e. higher ratio of humic-like substances to protein-like substances) with depth in the active layer. The elevated HIX values continue into the upper permafrost, but start decreasing again with greater depth (>~20 cm) within the permafrost. An explanation is be the presence of a transition layer and/or paleo-active layer. During the Holocene thermal maximum (HTM) (~10.6 cal ka. BP) active layer depths reached a depth of at least twice

the modern depth (Fritz et al., 2012; Fouché et al. 2020) implicating that the current permafrost may have been active layer in the past. This, together with interannual variability of maximum thaw depth explains why shallow permafrost bioavailability (and DOM composition in general) resembles that of the deep active layer more than anything (i.e. it has already undergone several periods processing and influx of soil water). More "untouched" bioavailable and less degraded OM is found deeper in the permafrost, where we had no incubation samples.

### 4.2.3 Bias through sampling depth and horizon limitations

Our sampling was limited by practical constraints in the field: permafrost cores from which we could extract sufficient pore water to perform incubation are from relatively shallow depths (~10 to 50 cm below permafrost



table, taken with a SIPRE corer), whereas smaller samples that were used for DOM spectral characterization reached greater depths (~10 to 100 cm below permafrost table, taken with a steel tube and sledgehammer). These deeper samples indicate increasing bioavailability with depth (e.g. fresh, low MW, low HIX, low aromatic DOM), however we could not confirm this with incubation experiments on the same samples.

### 4.2.4 Pre-incubation DOC losses

The spectral signatures of our DOM samples hint toward rapid processing upon thaw. As samples are thawed and prepared for porewater extraction it is inevitable that microbial activity starts up as well. The fact that we measure relatively low BDOC in permafrost DOM incubations but a clear signal of degraded DOM in samples at similar depths could mean that this permafrost DOM is indeed very labile. That would mean most of the BDOC is already lost in the first 24 hours before we extracted the water and is thereby missed by our measurements.

### 4.2.5 Translation of laboratory measurements to soil DOM dynamics in the field

It is difficult to accurately assess degradability and fate of DOM upon thaw, yet various studies have attempted tackling the problem of quantifying carbon fluxes from degrading permafrost landscape in in-situ (e.g. Schuur et al., 2009; Natali et al., 2014; Plaza et al., 2019), bulk soil incubation (e.g. Dutta et al., 2006; Lee et al., 2012; Gentsch et al., 2018) and lateral flux specific experiments (e.g. Kalbitz et al., 2003; Kawahigashi et al., 2006; Vonk et al., 2013, 2015). Porewater DOM incubation experiments as performed in this study are rare, however most confirm that DOM character and thus lability is highly variable on spatial scales (e.g. Shirokova et al., 2019; Fouché et al., 2020; MacDonald et al., 2021). Our results fall within the range of what is found in other studies (Vonk et al., 2015; Selvam et al., 2017) but are on average rather low for permafrost compared with active layer due to the reasons highlighted above. Nevertheless, our results show that DOM from, both, permafrost and active layer will likely be degraded within the soil. A relatively small proportion of the total SOM pool enters the aquatic system as DOM. Depending on transport times that DOM has undergone significant processing by the time it reaches the ponds, IWP troughs and headwater streams. Although small compared to the soil organic matter stock, the aquatic DOM 'stock' we observe in streams is predominantly terrestrially derived and high in aromatic, degraded components.



**4.3 Stream water OM dynamics and drivers during the warm season**

The OC export from our investigated watershed is dominated by DOC (mean$_{DOC}$ : mean$_{POC}$ = 16.03 mg L$^{-1}$ : 0.41

590     mg L$^{-1}$). This may be explained by the flat topography in this area which minimizes impacts of bank and thermo-
erosion that enhance sediment mobilization (Costard et al., 2003), along with long residence times of ground water
within the soil facilitating DOC leaching (Connolly et al., 2018). Similar dominances of DOC have been shown
elsewhere in the panarctic watershed, both in large and small rivers (e.g. Holmes et al., 2012; Fabre et al., 2019;
Coch et al., 2020) but this dominance may be even more pronounced for low-relief tundra plains. Based on satellite

imagery (WorldView-2, DigitalGlobe Inc., acquired on July 18, 2018) an estimated ~ 80 % to ~ 90 % of our
watershed consists of HCP terrain. The δ$^{13}$C-DOC and δ$^{13}$C-POC signatures from the stream water at the outlet
compared to what was found in porewaters suggest that DOC is predominantly derived from HCP terrestrial
sources (Mann et al., 2015) whereas POC likely stems from primary production of phytoplankton growth within
the stream network or ice wedge troughs (Tank et al., 2011) or fragments of sedge biomass transported into the

stream (Wooller et al., 2007). According to our source apportionment ~ 81 % - ~ 90 % is of terrestrial origin with
48 ±19 % of the DOM/DOC stemming from permafrost. The remainder (22 ±15 %) is most likely aquatic
DOM/DOC produced by primary production (fig. 8). The tributaries δ$^{13}$C-signal, showing signs of primary
production indicates that these small ice wedge trough streams are not necessarily important for transporting
terrestrial OM. Presumably terrestrial OM is rather transported towards the main channel via supra-permafrost

base flow, instead of via IWP troughs. Alternatively, terrestrial OM present in the IWP troughs may quickly be
decomposed and/or incorporated into primary production. Increased hydrologic connectivity of IWP troughs as
well as increased connectivity via active layer deepening in permafrost watersheds (Lafrenière et al., 2019; Evans
et al., 2020) may lead to higher input of terrestrial OM.

The exported OC at the outlet is variable in concentration and geochemical signature over time, with three
observable patterns: (i) diurnal variation of CDOM abundance and optical properties, (ii) short/storm-induced
peaks in POC and dips in DOC and (iii) seasonal decline of DOM export (fig. 3a, b right):

        i.      We observed a diurnal pattern in CDOM concentrations (range: ~ 20 % - 25 %) at the stream outlet
(fig. 3a right). This diurnal pattern can be explained by both temperature and light dependent
                variability in productivity as well as variations in ground ice melt contribution and



evapotranspiration induced flow effects, which are in turn temperature dependent (Spencer et al., 2008; Ruhala et al.; 2017). We observe that CDOM fluctuates synchronously with water temperature (i.e. peaks in water temperature correspond with peaks in CDOM). Similarly, peaks in Sr (~0.9) and FI (~1.5) correspond with lows in temperature while for HIX this pattern is inverted (fig. S9). This dynamic may be explained by the increasing importance of primary production with high temperatures, and decreasing importance of deeper baseflow when temperature decreases (i.e. flow becomes shallower and the signal less deeply terrestrial). The highest values of HIX are found around the permafrost table (presumably where baseflow takes place) (fig. S7) and a decrease in HIX at the outlet in sync with lowering temperatures supports the idea of freezing up from below.

ii. A storm event on the 16th and 17th of August resulted in drawdown of CDOM and DOC concentrations and a sharp spike in POC load (fig. 3a, b right). This shifted the DOC:POC ratio from an average of 50 to 5.1. We also observed an increase in pH (from ~7 to ~8) and spikes in EC (up to ~19000 μS cm$^{-1}$ compared to an average of ~900 μS cm$^{-1}$). The storm event was characterized by strong northwesterly winds which, given the shape and orientation of the lagoon are likely to have pushed water up the stream channel. Due to its proximity to Ptarmigan bay lagoon the autosampler and multi-parameter probe are likely to have recorded the inflow of lagoon water during storm springtide. We observed two peaks in POC export with two different terrestrial δ$^{13}$C-POC signals, the first one (~ -31 ‰ on the 16th) lasted only 6 hours and the second one (~ -27.5 ‰ on the 17th) lingered on for ca. 18 hours before going back to the background signal around -29 ‰. Compared to the average δ$^{13}$C-POC signal at the outlet, which tends to a more primary produced signal (-29.31 ±0.8 ‰ SD. table 3) this jump to -27.5 ‰ seems to be resulting from the input of storm-induced flushing of terrestrial POC, from overland flow, wind-driven bank erosion, and/or bottom disturbance in upstream lakes and ponds.

iii. Our sensor and sample data show a decreasing trend in both CDOM and DOC concentration during our sampling period (fig. 3a,b right, fig. S9). This is likely related to a gradual decline in temperature in the late summer as is illustrated by temperature records from nearby Herschel Island I (fig. S1) that show our 10-day monitoring period lies on the falling limb of the annual temperature curve. The decrease in concentrations over time can be caused by (i) a decrease in temperatures and solar irradiation toward the end of summer, leading to lower primary production in the aquatic network. Additionally, (ii) lower temperatures may decrease the efficiency of OM soil leaching (Whitworth



et al., 2014) over time. And finally, (iii) it may also be likely that the active layer is already starting to freeze-up from below, leading to lower availability of soil DOC. Several studies have looked at seasonal variability of DOC and DOM composition and concluded that antecedent winter DOC is flushed out during freshet, resulting in a DOC peak with relatively low Sr, FI and high SUVA$_{254}$, indicative of DOM coming from the organic surface layer. As the summer season progresses, a steady increase in Sr, and FI and decrease in SUVA$_{254}$ and DOC continuing up to the very end of the season was found uniformly in rivers across the Arctic, linked to the increasing contribution of deeper soil horizons via deepening of the active layer (Neff et al., 2006; Spencer et al., 2008, 2009a; Holmes et al., 2008, 2011). In this respect our study shows similar trends in DOC, SUVA$_{254}$, Sr, and FI at the outlet over the course of the monitoring period.

Our source-apportionment using $\delta^{13}$C-DOC, Sr and a254/a365 as tracers showed that a high proportion of DOC within the stream originates from permafrost DOC contributions (~48 %), outnumbering the DOC influx from the active layer (~31 %) and primary production within the stream (~21 %). This is in stark contrast to larger (Siberian) arctic rivers (Wild et al., 2019), where fluvial DOM fluxes stem predominantly from recent terrestrial primary production sources. Due to the small catchment size the contribution of permafrost OC is likely more evident in small streams and the hiatus between these and larger arctic rivers may indicate that permafrost DOC is likely degraded before it reaches larger rivers. Moreover, small streams like the one in this study drain a degrading continuous permafrost landscape exclusively, whereas large arctic rivers drain also non-permafrost or discontinuous permafrost terrain, disproportionally contributing to the riverine OM fluxes (e.g. Frey and McClelland, 2007). The spatial and temporal extent of terrestrial permafrost inputs into stream networks may likely expand upon increasing severity of meteorological extremes. For instance, Schwab et al. (2020) found aged DOC downstream in the Mackenzie River main stem, following a warm summer and the second warmest winter on record.

This study shows that permafrost-DOM related processes are most visible close to the terrestrial-aquatic interface, i.e. in the headwaters of arctic rivers. With our results we also show that variability herein is highly seasonal and weather driven as our measurements at the outlet show diurnal, storm event and seasonal trend patterns within the duration of our relatively short (10 day) field campaign. This emphasizes the need for high-resolution long-term measurements in order to fully understand the mechanisms at work in the (Arctic) carbon cycles. Important is the





notion that cascading effects and food web interactions resulting from permafrost release into arctic headwaters may be hard to detect, but may have large impacts on sensitive arctic ecosystems (e.g. Vonk et al., 2015a).

**4.4 First estimate of regional fluxes from small streams**

Small coastal tundra watersheds such as Black Creek, presented here, are abundant in and representative for the lowland regions of the Arctic continuous permafrost zone. Understanding the main controls on OM fluxes and linkages between soils and streams on the scale presented by us is an important step towards understanding panarctic carbon and nutrient fluxes. Our results show that small tundra streams have a pronounced terrestrial DOM signal and a fresher POM signature, but both are highly variable with time and mainly controlled by weather

events. The data presented here support the notion that base flow near the permafrost table feeds into small arctic streams and that during active layer deepening this results in the mobilization of potentially labile organic matter that has been freeze-locked in permafrost. Transport times through the soil are long (much longer than our 7-day incubations) and groundwater may be stagnant in some areas, dependent on precipitation or polygon type/drainage conditions. Hence, our data suggest that a large fraction of (labile) DOM may be utilized before reaching the

stream network. Uncertainties concerning the groundwater flow dynamics in permafrost regions make it extremely difficult to assess biogeochemical responses to hydrological shifts (Lafrenière & Lamoureux, 2019). The ratio of POC vs DOC in the total OC export, as well as the contributions of in stream production, active layer and permafrost are variable and our data suggest that this variability is mostly weather and hydrology driven. Due to their abundance and proximity to the Arctic ocean, small tundra streams have the potential to export large

quantities of terrestrial permafrost organic matter into coastal waters. Based on our sampling and measurements at the outlet (mean discharge of 67 ±51.1 L s$^{-1}$) and mean DOC and POC concentration of 16.0 ±3.2 mg L$^{-1}$ and 0.41 ±0.2 mg L$^{-1}$, respectively, together with a delineated catchment area of ~4 km$^2$, we are able to make first estimates of seasonal DOC (~0.03 ±0.021g DOC m$^{-2}$ d$^{-1}$) and POC fluxes (~0.0006 ±0.00065g POC m$^{-2}$ d$^{-1}$). In these flux estimates we accounted for the variability in POC concentration due to storms in our sampling period

(i.e. the mean POC concentration included the storm days). In contrast to DOC, POC fluxes are likely to be highly variable since they are impacted by summer storm activity which can vary between 2 and 21 storms per year in the southern Beaufort Sea region (Hudak & Young, 2002). Arctic-type storms are most prevalent in July and August and average at 8 storms per season. Assuming a duration of 1-2 days per storm this would lead to an additional export of ~0.010-0.020 g POC m$^{-2}$ annually. The period through which lateral transport of OM occurs

is dependent on the thaw season length and the rate at which the seasonal active layer deepens. To estimate



seasonal fluxes, we use an average thaw season duration of 87.7 consecutive frost-free days calculated over the period 1950-2013 from "The Climate Atlas of Canada" (version 2, July 10, 2019, https://climateatlas.ca). Using this average, we calculate that Black Creek watershed (~4 km$^2$) exports an average of 8.12 ±6.4 t DOC yr$^{-1}$ and 0.21 ±0.2 t POC yr$^{-1}$. We calculated the entire Canadian Yukon Coastal Plain drainage area to be ~17000 km$^2$.

This yields an approximate 34.51 ±2.7 kt DOC yr$^{-1}$ and 8.93 ±8.5 kt POC yr$^{-1}$ for fluvial export. This preliminary scaling assumes that all streams have equally high loads, which is probably an overestimation. For example, larger rivers present in the area are likely to have lower DOC and POC loads as they drain areas with lower OM rich soils and longer transport time and distance allows for more OM processing. With longer, warmer seasons and higher storm frequencies which are predicted for the Arctic (Day et al., 2018), mobilization and export of POC

and DOC towards the Arctic Ocean may substantially increase.

## 4.5 Implications and future research

This study focuses on data retrieved during the latest stage of the thawing season, when active layer depths are at their seasonal maximum. Our results show that OC quality and quantity varies between different soil horizons and landform (i.e. polygon type). Evolution of landforms via degradation of IWP's from LCP to HCP is likely to result

in increased drainage and drying of the landscape and increased net runoff as a consequence. Our results show that the balance between lateral and vertical flux is therefore likely to shift toward lateral flux as mobilizable OM will be flushed out of the system more effectively. The shift towards drainage and export rather than within-ecosystem-processing may have strong effects on the arctic lowland tundra biodiversity and food web interactions since these are to a large extent based on wetland-ecosystems (e.g. Vonk et al., 2015a; Liljedahl et al., 2016).


In parallel to annual active layer depth deepening, older and potentially more labile OM pools will become available for degradation. In this study we find indications that the most labile fractions are utilized within the soil, and we expect that even in a more well-drained HCP system, residence times within the soil will be long enough to allow for the utilization of the majority of labile DOM. The results show that concentrations of DOC

in permafrost are much higher than in the active layer. This together with the notion that labile DOM would be converted quickly leads to the expectation that under current climate trends, small arctic catchments affected by permafrost degradation may export higher loads of recalcitrant DOM. This in turn could impact aquatic food webs in various ways. A recent study by Wologo et al. (2021) suggests that negative priming effects occur as a result of influx of bioavailable permafrost DOM. However, our results show influx of less bioavailable permafrost hence

we deem negative priming unlikely. Due to its strong coloration permafrost and deep active layer derived CDOM
        could significantly impact light dependent processes in the aquatic network. Moreover, chemical composition of
        permafrost and deep active layer DOM (e.g. high values of ammonium were found) may impact aquatic
        biogeochemistry. More research is needed to elucidate these ecosystem interactions.

To better understand and implement lateral permafrost-OM-dynamics into climate models, more quantitative and
        qualitative data on the distribution and behavior of small, pan-arctic permafrost catchments is needed. Moreover,
        there is a simple need to map these watersheds at the basis of the aforementioned challenge. Both, the former and
        latter could be achieved by more long-term monitoring on a larger spatial scale, e.g. by installing sensors and
        conducting repetitive field research in designated representative areas as well as by aggregating databases with
field and remote sensing data. Lastly, by focusing on optical properties of DOM it is relatively easy and cost-
        effective to trace changes in watershed biogeochemistry, as optical measurement techniques are uncomplicated
        and readily available. These techniques together with standardization of methods are there for recommended for
        a harmonized approach on understanding lateral permafrost-OM-dynamics.

## 5 Conclusions

This study investigates the lateral release of organic matter in an arctic lowland IWP tundra watershed, subject to
        permafrost degradation. Soil porewater DOM properties and DOC concentrations in the Black Creek catchment
        vary between thermal layer (i.e. active layer and permafrost) and landform (i.e. LCP and HCP), reflecting
        differences in drainage patterns and waterlogged conditions. Also, within the active layer, DOM signatures vary
        between polygon types due to differences in drainage status (i.e. LCP is more water logged, HCP well drained).
HCP active layers show a more degraded OM signature. When further arctic warming transitions LCP landscapes
        into HCP-dominated settings, this may lead to an increasing flux of degraded DOM from soils to streams.

        Dissolved carbon yields (mg DOC g$^{-1}$ soil OC) increase with soil depth, yet show a larger variability around the
        permafrost table. Gleyed soil samples from mineral horizons had relatively high dissolved yields while having
low SOC contents, hence accumulation of DOM from other horizons in these gleyed horizons is likely. Porewater
        incubation experiments show 5-17 % DOC loss after 7 days, with higher losses for active layer than permafrost.
        The incubated permafrost samples are mostly from within the transition layer where degradation has likely

occurred in the past. Optical properties however indicate increasingly fresh and (potentially) labile OM with depth.
Long transport time of porewater DOC within these low-relief catchments suggest that most permafrost DOM is
processed/degraded within the soil before it reaches the stream network.

Black Creek transports much more DOC than POC, but storm events change that ratio by an order of magnitude.
Our 10-day monitoring period shows diurnal, weather-driven, and long-term patterns in OC concentrations and
properties. Source apportionment of stream DOC using $\delta^{13}C$ and DOM-spectral signatures show a dominance of
terrestrial OC over autochthonous production, and a deep active layer/permafrost-DOC contribution around 48
%. This contrasts with larger arctic fluvial systems that are dominated by recent terrestrial production. First
upscaling estimates of annual Black Creek fluxes give values of 8.12 ±6.4 t DOC yr$^{-1}$ and 0.21 ±0.2 t POC yr$^{-1}$.
Rough upscaling to the entire Canadian Yukon Coastal Plain drainage area (~17000 km$^2$) yields an approximate
34.51 ±2.7 kt DOC yr$^{-1}$ and 8.93 ±8.5 kt POC yr$^{-1}$ by fluvial export.


High frequency measurements at the outlet in combination with in-situ weather observations underline the highly
variable nature of small arctic watersheds and their susceptibility to changes. To get a more thorough
understanding of arctic watersheds and their responses to climate change and permafrost degradation, it is
important that more geographically widespread and longer timespan covering monitoring efforts of these streams
are implemented, e.g. through sensor installations, use of cost-effective optical proxies to monitor change. Further,
combining remote sensing data with field observations and machine learning techniques pose a powerful tool for
upscaling.

**Acknowledgements**

We thank all those who have made contributions that have led to this publication. We thank the Yukon Territorial
Government, Yukon Parks (Herschel Island Qikiqtaryuk Territorial Park), and the Aurora Research Institute for
their support during this project. The work presented here was done under the Nunataryuk project, which received
funding under the European Union's Horizon 2020 Research and Innovation Program under Grant Agreement
773421. We wish to express our special gratitude to C. Stedmon for providing laboratory access, equipment and
guidance. And to S. Verdegaal-Warmerdam, A. Dalhoff Bruhn Jensen, C. Burau, J. Gimsa, S. Stettner, A.
Beamish, K. Klein, R. Broekman, R. van Logtestijn, M. Sanchez Roman, M. Fritz and L. Bröder for laboratory



analysis, and field and laboratory assistance and conceptualization of the research project. We thank S. McLeod, P. Archie and F. Dillon for their helpful insights and support in the field.




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

1025





# Main Figures

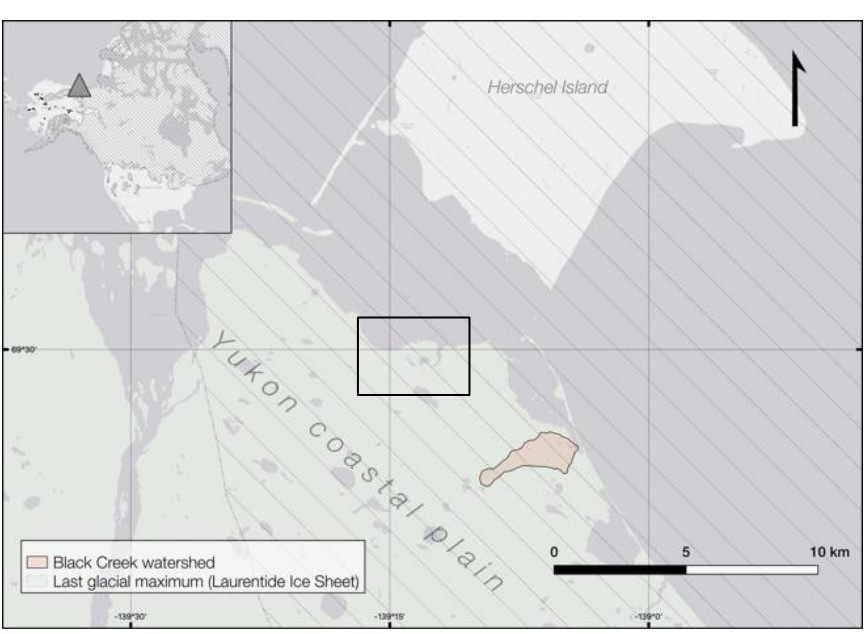

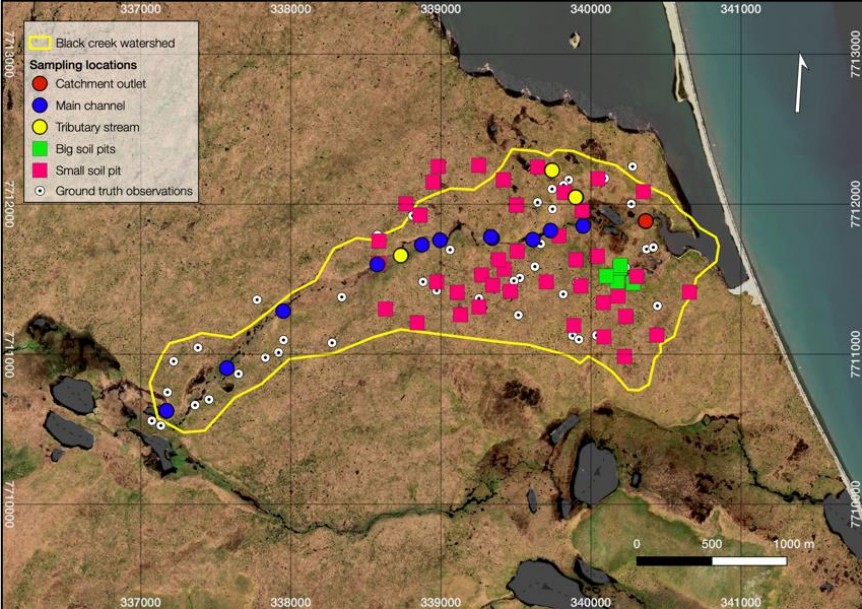

**Figure 1. Location of Black Creek watershed on the Yukon coastal plain (upper panel) and detailed catchment image showing the different sampling locations and sampling types (lower panel) (satellite imagery: WorldView-2, DigitalGlobe Inc., acquired on July 18, 2018).**




# Main Figures

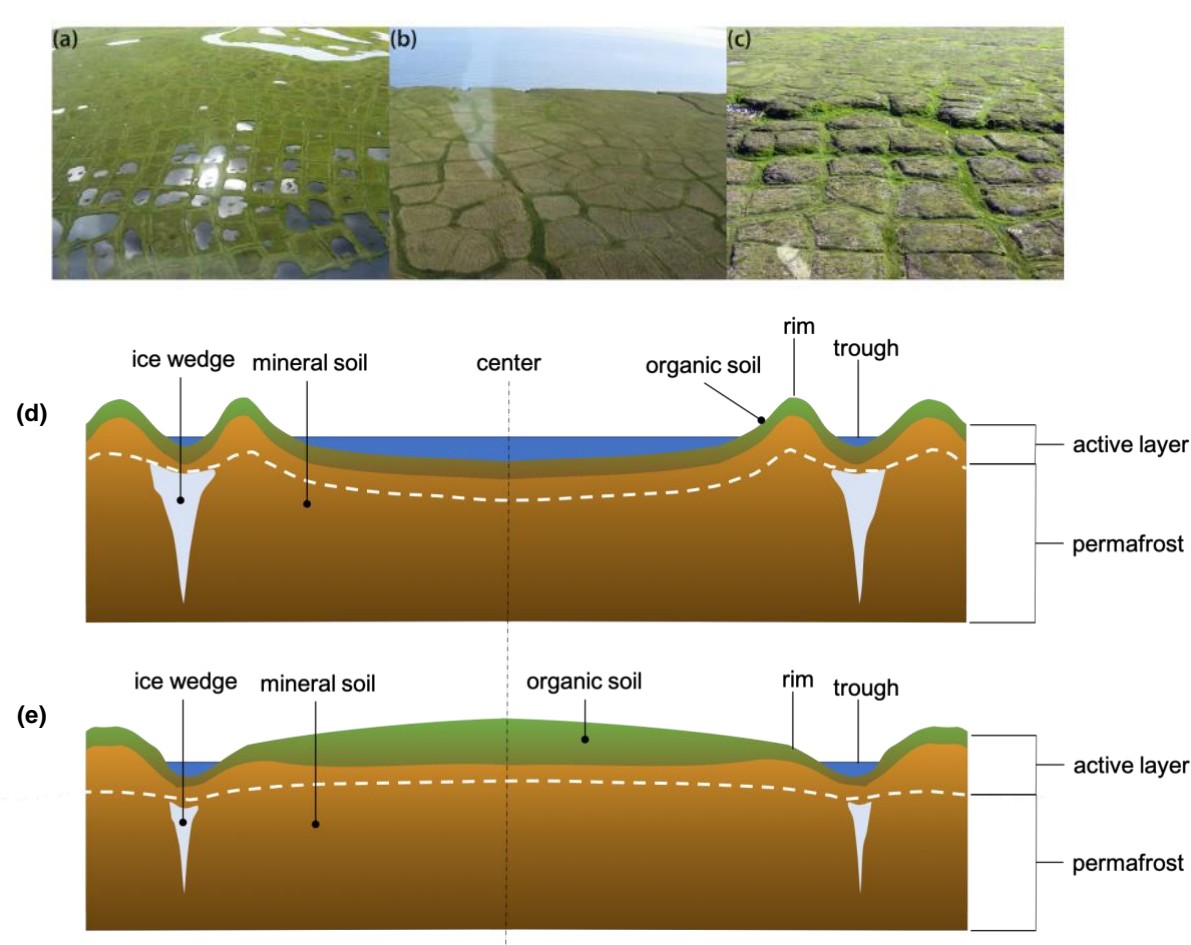

Figure 2. Examples of typical low (a), flat (b) and high (c) centered polygons as seen along the Yukon coast (adapted from Fritz et al., 2016). Schematic of a low centered polygon (d) and high centered polygon (e).





# Main Figures

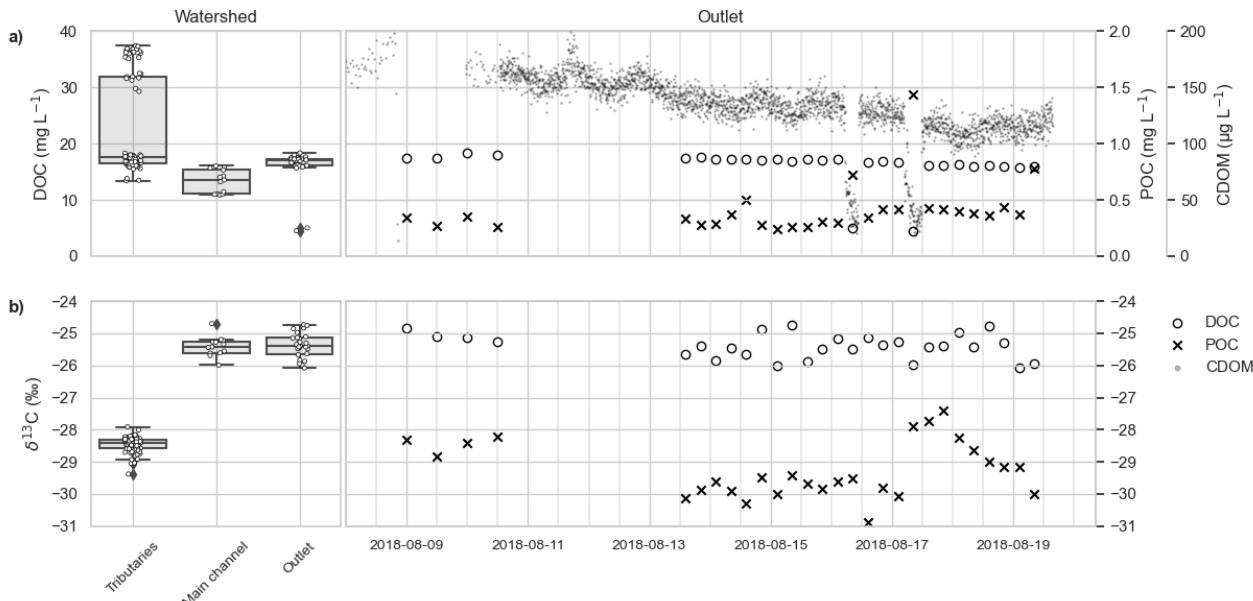

**Figure 3. DOC concentrations (mg/L) measured within tributaries, main channel and outlet (panel A, left), and POC (mg/L; crosses), DOC (mg/L; circles; note scale is on left panel) and cDOM (µg/L; filled small circles) concentrations within stream water at the watershed outlet over time (panel A, right). $\delta^{13}$C-DOC isotopic signal within tributaries, main channel and outlet (panel B, left), and $\delta^{13}$C-DOC and $\delta^{13}$C-POC over time at the catchment outlet (panel B, right). Note that two clear drops in the CDOM measurements on the 16th and 17th of August mark a storm event. This is also visible in the $\delta^{13}$C-POC and to lesser extent $\delta^{13}$C-DOC source shift (B, right) around these dates.**






# Main Figures

**Figure 4. Porewater DOC concentrations (mg/L) across the entire watershed and identified polygon types (HCP, LCP, flat) for permafrost (dark grey box plots) and active layer (light grey) (a), versus depth (b), yield of DOC (in mg) per gram dry soil (c) and yield of DOC (in mg) per gram carbon (d). Color indicates soil organic carbon content (%). Marker type distinguishes soil horizon. Cryoturbated soil samples are annotated with '*' and gleyed soil samples with 'g'.**





# Main Figures

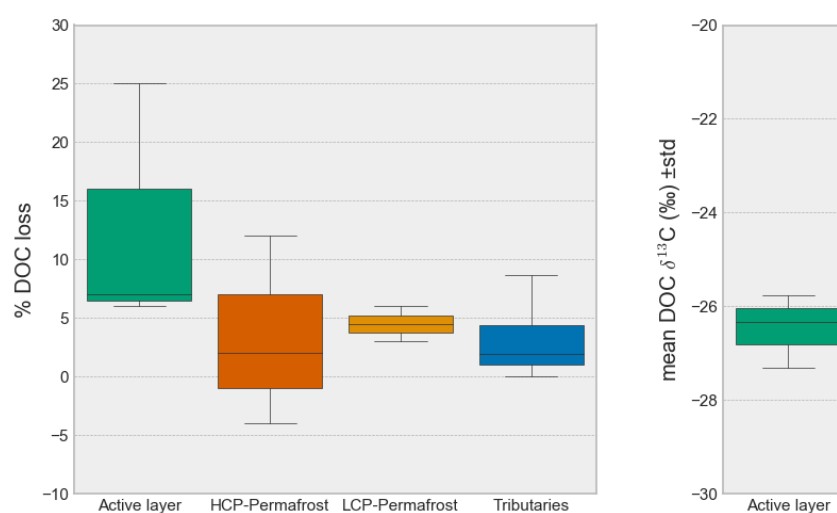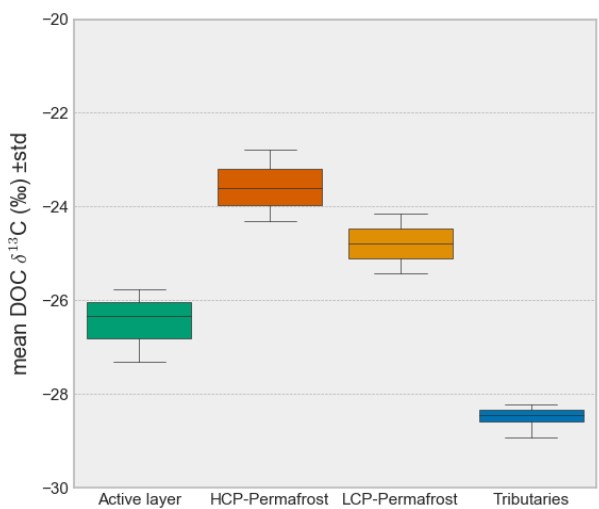

**Figure 5. Loss of DOC (%) after 7 days incubation (left), and initial δ¹³C-DOC (right) for the sources active layer (porewater), HCP and LCP permafrost (porewater) and tributaries (stream water).**



# Main Figures

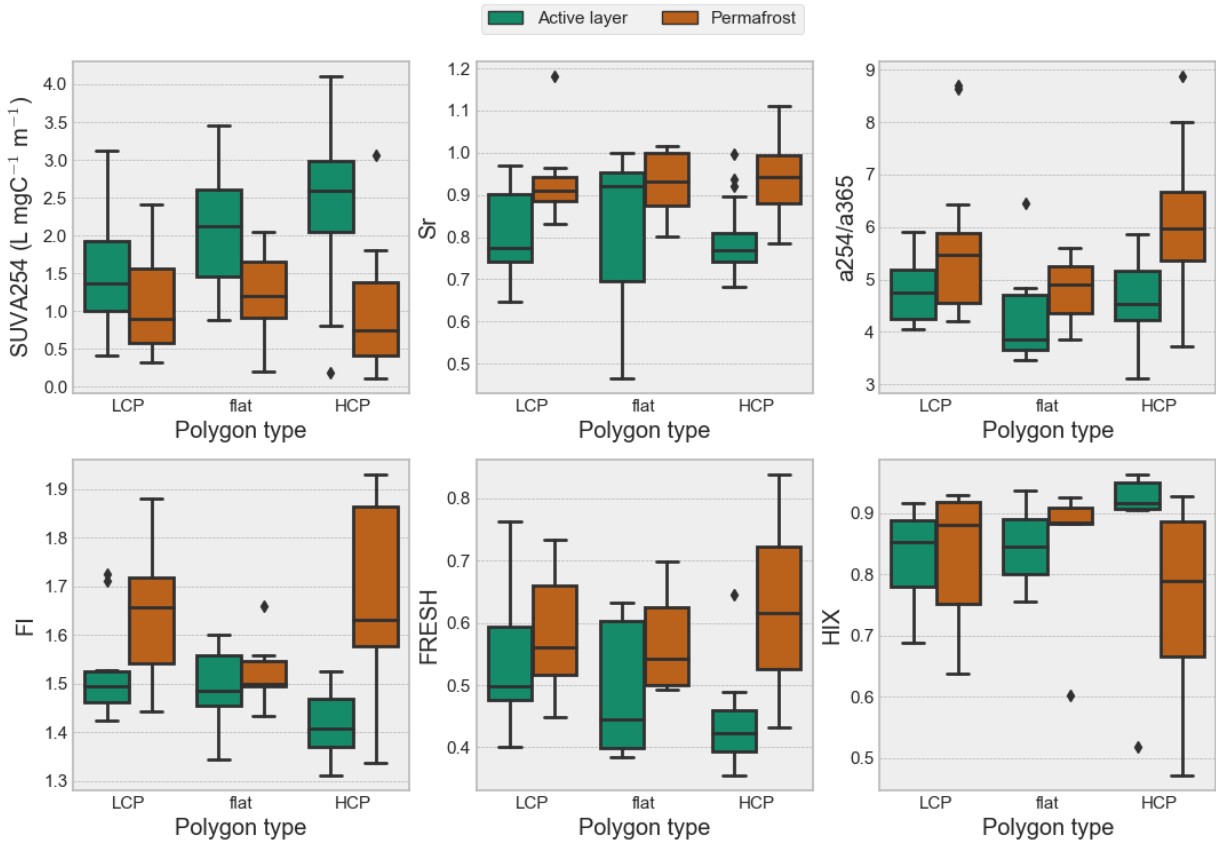

**Figure 6. SUVA$_{254}$, slope ratio (Sr), absorbance ratio (a254/a365), fluorescence index (FI), freshness index (FRESH) and humification index (HIX) for both thermal layers in each polygon type. Differences between the two thermal layers are largest in HCP and smallest in LCP except for FI where flat type polygon has the smallest difference between active layer and permafrost. Indicating a shift in biogeochemical processing of DOM as IWP degradation progresses (i.e. transition from LCP to HCP).**






# Main Figures


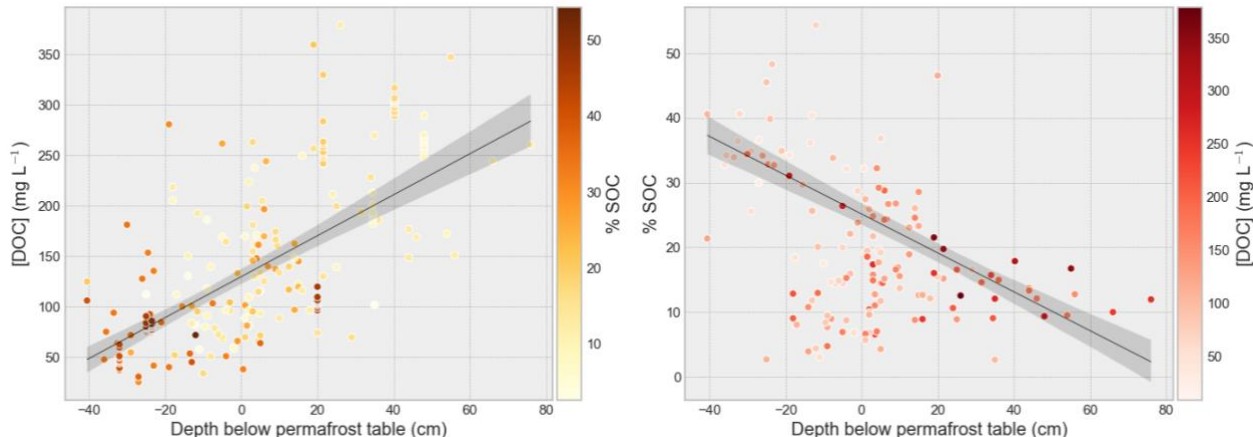

**Figure 7. DOC concentration (mg L⁻¹) vs depth to the permafrost table (cm; left panel) and SOC content (% of dry weight) vs depth to the permafrost table (cm; right panel), where negative indicates samples within the active layer and positive below the permafrost table. Colors of the dots show SOC % and DOC**
**concentration respectively, and show the inverse correlation between soil organic carbon content and DOC concentration over the depth of the soil profile. We observe elevated DOC concentrations corresponding with low SOC content around the permafrost table, which may be an effect of flushing in from the overlying Oi-horizons and accumulation above the permafrost table. Linear regression of DOC with depth respective to permafrost table in cm (i.e. active layer depth $Z_{AL}$) yields $[DOC] = 2.028 \cdot Z_{AL} +$**
**$134.60$, R² = 0.71, p = 8.6e⁻³⁷, for $\%SOC = -0.3 \cdot Z_{AL} + 25.15$, R² = 0.55, p = 1.04e⁻¹⁹.**





# Main Figures

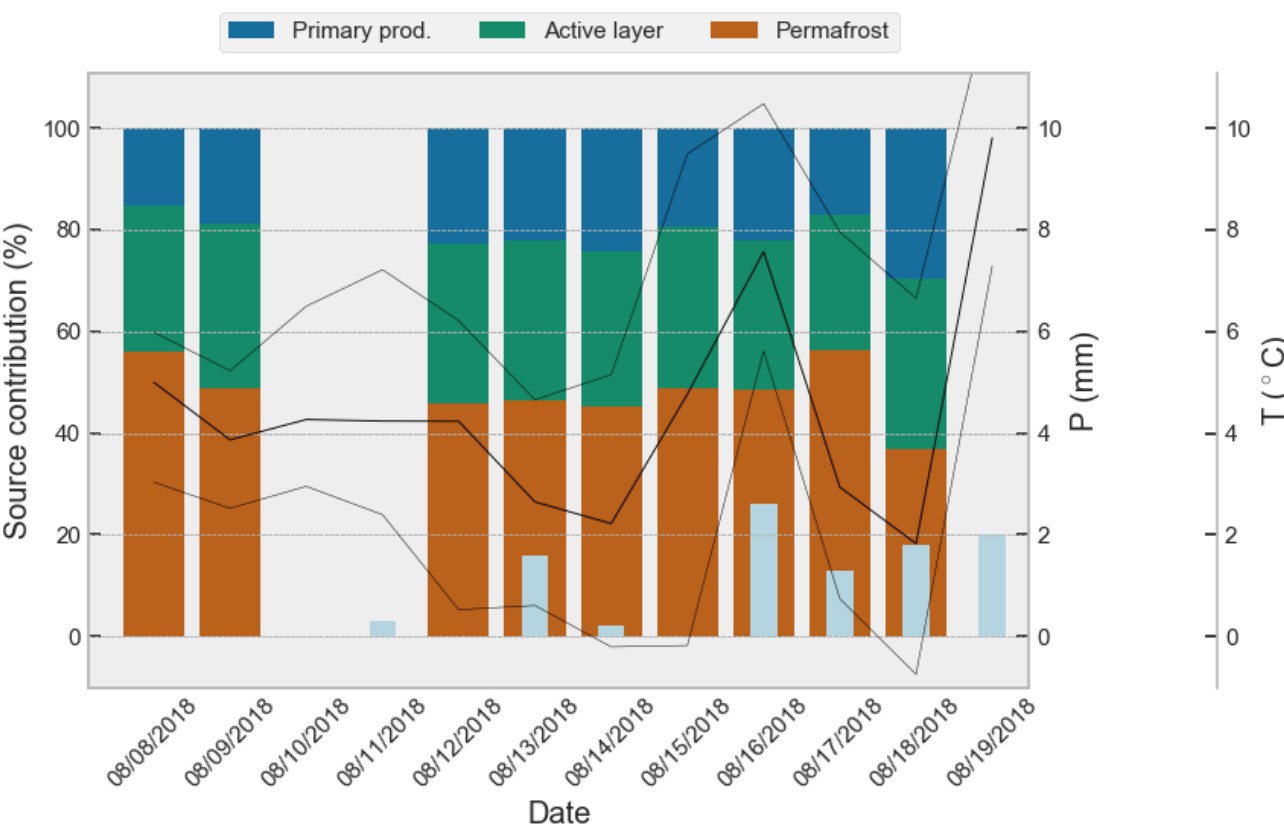

**Figure 8. Modeled DOM source contributions, aggregated by day, to catchment outlet sample over time. Plotted together with recorded rainfall (light blue bars) and recorded minimum, maximum and mean air temperature (black lines)**




# Main Tables



# Main Tables

| Category | Parameter | DOM Indicator | Method |
|---|---|---|---|
| **Absorbance** | Absorption coefficients ($\alpha_{350}$) [m$^{-1}$] | CDOM content | Absorption coefficient at wavelength 350 nm |
| | Absorption ratio | Tracing relative changes in DOM molecular weight (De Haan & De Boer, 1987) | Ratio of absorption coefficients $\alpha_{254}$:$\alpha_{350}$ [-] |
| | Specific ultraviolet absorbance (SUVA$_{254}$) [L mg$^{-1}$ m$^{-1}$] | Aromaticity, $\Delta^{14}$C of hydrophobic organic acid (HPOA) fraction of DOC (O'Donnell et al., 2014), $\Delta^{14}$C-DOC (Butman et. al., 2012) | $SUVA_{254} = \alpha_{254}/[DOC] \cdot 100$ (Weishaar et al., 2003) |
| | Spectral slope ($S_{275-295}$, $S_{350-400}$) [nm$^{-1}$] and Slope ratio ($S_R$) [-] | Molecular weight | Nonlinear fit through absorption coefficients between 275-295 nm ($S_{275-295}$), 350-400 nm ($S_{350-400}$) and $S_R = S_{275-295}:S_{350-400}$ |
| **Fluorescence** | Humification index (HIX) | Indicator of degree of humification or humic substance content (Fellman et al., 2010, Fouché et al., 2017) | The area under the emission spectra 435–480 nm divided by the peak area 300–345 nm + 435–480 nm, at excitation wavelength 254 nm (Ohno, 2002) |
| | Fluorescence index (FI) | Identify relative contribution of terrestrial vs microbial sources (Fouché et al., 2017) | The ratio of emission intensity at wavelength 470 nm and 520 nm, at excitation wavelength 370 nm (McKnight et al., 2001, Cory et al., 2010) |
| | Freshness index ($\beta$:$\alpha$) | Higher values represent higher proportion of fresh DOM (Fouché et al., 2017) | Emission intensity at 380 nm divided by the maximum emission intensity between 420 nm and 435 nm at excitation 310 nm (Parlanti et al., 2000, Xenopoulos, 2009) |

**Table 1. Overview of spectral (absorbance and fluorescence) indices used for DOM qualification.**





# Main Tables

**Table 2. Stable water isotope values mean and standard deviation by source.**

| Source | δ²H (*mean, std*) [‰]) | | δ¹⁸O (*mean, std*) [‰]) | | d-excess (*mean, std*) [‰]) | |
|---|---|---|---|---|---|---|
| Permafrost porewater (n=18) | -123.3 | ±7.2 | -15.3 | ±1.5 | 1.1 | ±7.6 |
| Active layer porewater (n=8) | -122.6 | ±3.3 | -15.9 | ±0.7 | 4.6 | ±3.3 |
| Tributaries (n=10) | -124.0 | ±2.8 | -16.0 | ±0.4 | 3.9 | ±1.0 |
| Main channel (n=30) | -123.5 | ±8.3 | -15.8 | ±1.1 | 2.5 | ±0.9 |

**Table 3. $\delta^{13}C$ values of DOC and the POC/SOC for various sources in the catchment. *Note that LCP-active layer $\delta^{13}C$ -DOC was calculated from the linear relationship between available HCP-active layer $\delta^{13}C$ -DOC and $\delta^{13}C$ -SOC and using the $\delta^{13}C$-SOC of LCP-active layer. **Values listed**
**are SOC for HCP/LCP active layer and permafrost, and POC for tributaries and main channel.**

| Source | $\delta^{13}C$-DOC (mean ±std) | $\delta^{13}C$-SOC/POC** (mean ±std) | $\delta^{13}C$ difference |
|---|---|---|---|
| HCP-permafrost | -23.68 ±1.2‰ | -27.35 ±0.8‰ | +3.67‰ |
| LCP-permafrost | -25.05 ±0.5‰ | -27.29 ±1.0‰ | +2.25‰ |
| LCP-active layer* | -26.71 ±1.1‰ | -28.27 ±1.0‰ | +1.56‰ |
| HCP-active layer | -26.38 ±1.1‰ | -27.58 ±0.8‰ | +1.21‰ |
| Tributaries | -28.48 ±0.2‰ | -32.68 ±2.0‰ | +4.21‰ |
| Main channel | -25.40 ±0.4‰ | -29.31 ±0.8‰ | +3.91‰ |

**Table 4. Change of mean $\delta^{13}C$-DOC (‰ VPDB) between active layer, permafrost for the two polygon types (LCP and HCP).**

| Source | $\delta^{13}C$-DOC at $T_0$ | $\delta^{13}C$-DOC at $T_7$ | $\delta^{13}C$-DOC at $T_{21}$ | mean change |
|---|---|---|---|---|
| Active layer (Oi) (n=3) | -26.38 ±1.1‰ | -26.47 ±0.7‰ | -26.22 ±0.8‰ | +0.16‰ |
| LCP-permafrost (n=2) | -25.05 ±0.5‰ | -24.78 ±0.7‰ | -24.89 ±0.6‰ | +0.16‰ |
| HCP-permafrost (n=3) | -23.68 ±1.2‰ | -23.57 ±0.7‰ | -23.59 ±0.8‰ | +0.10‰ |





# Main Tables

**Table 5. Summary of fluorescence and absorbance indicators of DOM quality. Significant differences and sample sizes are indicated in table S4.**

| Index | HCP | | LCP | | Flat | | Streams | |
|---|---|---|---|---|---|---|---|---|
| *mean ±std* | *Active layer* | *Permafrost* | *Active layer* | *Permafrost* | *Active layer* | *Permafrost* | *Tributaries* | *Main channel* |
| **FI** | 1.42 ±0.1 | 1.68 ±0.2 | 1.53 ±0.1 | 1.64 ±0.1 | 1.49 ±0.1 | 1.52 ±0.1 | 1.47 ±0.03 | 1.49 ±0.02 |
| **HIX** | 0.86 ±0.2 | 0.75 ±0.1 | 0.83 ±0.1 | 0.83 ±0.1 | 0.85 ±0.1 | 0.84 ±0.1 | 0.96 ±0.02 | 0.94 ±0.01 |
| **BIX** | 0.44 ±0.1 | 0.64 ±0.1 | 0.54 ±0.1 | 0.60 ±0.1 | 0.50 ±0.1 | 0.57 ±0.1 | 0.50 ±0.04 | 0.56 ±0.04 |
| **FRESH** | 0.43 ±0.1 | 0.62 ±0.1 | 0.54 ±0.1 | 0.59 ±0.1 | 0.49 ±0.1 | 0.57 ±0.1 | 0.50 ±0.04 | 0.56 ±0.04 |
| **a254/a365** | 4.66 ±0.7 | 6.08 ±1.1 | 4.81 ±0.6 | 5.67 ±1.4 | 4.33 ±1.4 | 4.80 ±0.6 | 5.30 ±0.2 | 5.55 ±0.3 |
| **Sr** | 0.79 ±0.1 | 0.94 ±0.1 | 0.81 ±0.1 | 0.92 ±0.1 | 0.81 ±0.2 | 0.93 ±0.1 | 0.80 ±0.02 | 0.86 ±0.03 |
| **SUVA$_{254}$** | 2.42 ±0.9 | 0.91 ±0.7 | 1.50 ±0.8 | 1.10 ±0.7 | 2.09 ±0.9 | 1.23 ±0.6 | 3.86 ±1.8 | 3.90 ±2.2 |

**Table 6. Mean relative contribution of three identified sources (source fractions) to the integrated signal at the catchment outlet, using $\delta^{13}$C-DOC for the primary production endmember.**

| Source | 2.5% percentile | median | 97.5% percentile | mean | Std. |
|---|---|---|---|---|---|
| Permafrost | 0.075 | 0.488 | 0.830 | 0.479 | 0.19 |
| Active layer | 0.013 | 0.271 | 0.788 | 0.305 | 0.21 |
| Primary production | 0.015 | 0.192 | 0.561 | 0.216 | 0.15 |

**Table 7. Mean relative contribution of three identified sources (source fractions) to the integrated signal at the catchment outlet, using $\delta^{13}$C-POC for the primary production endmember.**

| Source | 2.5% percentile | median | 97.5% percentile | mean | Std. |
|---|---|---|---|---|---|
| Permafrost | 0.124 | 0.564 | 0.872 | 0.545 | 0.19 |
| Active layer | 0.025 | 0.315 | 0.808 | 0.341 | 0.22 |
| Primary production | 0.006 | 0.095 | 0.329 | 0.114 | 0.09 |