# Peer review of "Dissolved organic matter characterization in soils and streams in a small coastal low-arctic catchment"

_Biogeosciences, 2021_

## Author Response (AR1)

**REVIEWER 1.**

**Specific Comments:**

The EC at the outlet is rather high (954 uS/cm) is there tidal influence during base flow? Perhaps this is a typo? Looking at supplementary values they are around 90-100's uS/cm, maybe the authors meant a mean of 95.4? Otherwise, if this is correct, was salinity taken into account in terms of DOM behavior at the outlet? As well as comparison between tributaries, main channel, and the outlet?

*This is correct, we have taken the mean over the entire monitoring period, including the storm event induced EC peaks. We do not think there is tidal influence during baseflow, but it seems there is during storm events. We will add a line stating the mean EC ±std when omitting these peaks. The mean EC value then becomes $100 \pm 14.0 \mu S \ cm^{-1}$.*

Could you offer an explanation for why DOC dilutes but POC increases during storms (line 365)? Could this be weaved into the discussions of DOM sources in the discussion? POC is not mentioned very much further on, although I am cognizant that it's a small contribution to the DOM pool.

*Thank you for your question, I will further clarify my interpretation of this phenomenon in the discussion as follows: The release of DOC seems to be fairly consistent and unaffected by storm events, hence adding water to the system in the form of precipitation basically dilutes the DOC concentration. For POC however this is a different story: The low-energy system that is this coastal tundra plain catchment mobilizes low quantities of POC during baseflow conditions. Presumably this low-flow POC consists primarily of relatively fresh in-stream production. During storm flow, a combination of increased runoff and wind-driven stirring of the water column of ponds, lakes and streams mobilizes POC that is otherwise not in suspension. Hence the peak in POC concentration during stormflow. Also this POC showed a more enriched $\delta^{13}C$ value, pointing towards the influx and/or suspension of more terrestrial OC.*

In lines 507-509, you mention DOM flushing with runoff, are POC values greater in these sites? IF the data is available, perhaps they could be mentioned? Could be another piece of evidence of this and role of storms on this watershed, especially in the switch of DOC:POC ratios.

*We do not have data supporting this unfortunately. Moreover, it seems that POC flushing from soils in this low-relief catchment is not dominant. Hence, the aforementioned statement is directed at the flushing of leached DOM from the soil and/or degradation of leached DOM within the soil. Since HCP is generally more well drained we suggest that degradation of DOM (better drainage = better aeration) as well as flushing of DOM is more promoted in HCP.*

Figure 4a and line 384 - Perhaps add a letter map to show these differences or no difference? I know the figure is already busy, but these details will help orient the reader when looking at the figures. It's really hard to differentiate between permafrost and active layer boxplots. Is it possible to change the fill of the boxplots to make this more clear? Maybe a translucent green and orange like in other figures. Or any slight change to make the boxplot fill pop out a bit more. And while on Figure 4, the "*" and "g" are really hard to see.

*Thank you for your note, we changed the color and visibility of 4a. The \* and g are not easily improved, we will colorize them in a bright color to make them more visible.*

I wonder if the very high SUVA value should be removed and mentioned parenthetically? This might help to make the section starting in 425 a bit more fluid and clearer. A SUVA value that high is also odd so perhaps take away some of the attention given to it.

*Yes, thank you that is a good point. We will remove the mention of the high SUVA$_{254}$ value.*

Section 4.2 header is about the mobilization of OM from soils to streams yet most of the subsections are of the soil columns processes. Lateral exports of inputs to streams are not mentioned until the very last paragraph. Is it possible to elaborate a bit more on the terrestrial aquatic linkages in the subsections? Or the contributions from HCP and LCP to the stream? Where possible. Or change the header and remove "to streams".

*That is a valid point, thank you. While we do think that aquatic DOC processing already starts within the soil (so that is where the terrestrial-aquatic linkage/journey begins) we agree that there is little stream focus here and have removed the "to streams" from the header.*

In line 580, could you briefly mention the range of other studies in comparison to yours, just to help put think into context and a quick refresher for the reader.

*This makes sense, we have added "(0 – 67% BDOC) (Vonk et al., 2015)".*

Have you considered the role of photodegradation in the temporal declines of CDOM and DOC (pint iii in section 4.3)? This might be another aspect of the OM dynamics in addition to changes in temperature?

*We have considered it, but think that it is unlikely this relates to the gradual decline in CDOM/DOC because (i) in August the average hours of daylight are slowly declining, and (ii) there was more rain/cloudy weather towards the second half of our monitoring period.*

**Technical Corrections:**

Add a comma between layer and primary in line 466

*Thank you for your comment, we have added the comma.*

Line 544-545, the wording here is confusing, some clarification is needed on what is meant by "leaches less than more degraded"

*Thank you for pointing this out, we have rephrased: "We interpret this as a high abundance of fresh plant material, which releases less DOM per unit time than more degraded plant material (i.e. living plant material has a lower leaching potential)".*

Line 553, remove "be" after "is".

*Done*

Add a comma between "event" and "and" line 674

*Done*

Figure 3, a minor detail, in panel a, are POC and DOC on the same axis? Is it possible to this to the axis label for clarification?

*This is a bit unclear indeed but because we didn't want to repeat axis for DOC on the right side it is only plotted on the left-hand side of the plot. Hence the [DOC] axis is continuing from the plot on the left to the plot on the right in the top panel a. The same holds for $\delta^{13}C$ in panel b. We will clarify this in the caption.*

**REVIEWER 2.**

**General Comments**

This paper presents a robust dataset including field and laboratory observations to characterize organic matter dynamics and drivers of these dynamics in a small, coastal Arctic watershed underlain by permafrost. The dataset presented is a novel contribution to the science and it is clear that significant thought went into its collection and analyses to infer as much as possible about organic matter cycling from soils to the streams, including distinct types of tundra plain. While this is a novel, much-needed dataset to further inform the community on these understudied systems, significant changes are needed to make this paper suitable for publication.

Most of the material is relevant to the paper, but there is a lack of flow. A general improvement could be to introduce what is known, then highlight what is not known. For example, it is unclear what I should take away from the second last paragraph in the Introduction. What factors are specifically driving terrestrial-aquatic DOM dynamics in regions underlain by permafrost? A range of factors are provided, some specific and some not, but a cohesive framing of these dynamics is not provided. Under "historic" conditions, how would we characterize DOM flux from permafrost soils to streams? Then, under disturbed conditions, how would we characterize DOM flux from permafrost soils to streams? What do we know, and what are we still unsure of? This framing across all paragraphs should follow a relatively straight line that leads us to the purpose of your study.

The introduction would benefit from more background on IWP tundra and processes unique to these systems that impact OM cycling. The Introduction would benefit from reorganization, focusing on broad scale impacts of climate change on Arctic systems and impacts on OM cycling, then transitioning to IWP processes and OM cycling. Within this context, the reader would be more clearly led to the purpose of the study introduced in the final paragraph. Overall, it is difficult throughout the paper to understand the purpose.

*We agree, upon reassessment, that the introduction does not build up nicely toward the purpose of the study and that it would benefit from restructuring and improving the flow. In the re-written introduction we will:*

- *Follow the order suggested by the reviewer to be "broad scale impacts of climate change on Arctic systems and impacts on OM cycling, then transitioning to IWP processes and OM cycling", in which we will clarify what is known, and where knowledge gaps still exist*
- *As suggested, we will provide further background on IWP tundra and its unique processes.*
- *In the final paragraph specify our overall aim a bit better by stating the purpose; to gain a better understanding of the functioning of a degrading IWP catchment with respect to lateral OM transport in order to better assess the role of small arctic watersheds in our understanding of land-ocean OM budgets but will keep our four primary objectives, as these four objectives also central in the discussion.*

Alongside this need for re-organization and focusing, the purpose of this study needs to be emphasized and mirrored in the Discussion and Conclusion.

*Thank you. We currently have the headers in our discussion (4.1 through 4.4) agreeing with the 4 specific objectives that are mentioned at the end of the introduction. We also have shortened 4.2 and 4.4 to improve focus, and have incorporated a few more "topic sentences" at the beginning of paragraphs or sections to provide more direction and strengthen the link with the main objectives.*

There is a recurrence of using this dataset – collected over a 10-day timeframe in a single watershed – to better understand broader dynamics occurring in the Arctic. However, there is not enough spatial or temporal coverage to justify this extrapolation.

*We acknowledge the limited spatial and temporal extent of the data collected here. However, the time and location of sampling are deemed representative of river systems draining the Yukon coastal plain during thawing season. Moreover, there is hardly any data available that is detailed enough to make similar estimates. Hence, despite the high uncertainty we still value to present this calculated flux estimate even though it is a coarse attempt towards regional upscaling. We will however emphasize this uncertainty a bit more to be more transparent.*

There is significant time spent on in-stream primary production. If the dataset presented here clearly defines this contribution, that needs to be elaborated far more. In my opinion, there is not enough data here to draw clear conclusions on the role of primary production, as it doesn't appear this was directly observed.

*We acknowledge that taking our value of -28.48 ±1.0 ‰ (n = 9) based on $\delta^{13}C$-DOC values of the tributaries where we observed primary production (algal mats) is based on a limited amount of data. We also acknowledge that taking $\delta^{13}C$-DOC rather than $\delta^{13}C$-POC is complicating the endmember isolation because of mixing with water from other sources and/or isotope shifts during leaching of DOC from POC. However, our values are supported by other studies, for example Winterfeld et al., (2014) who used riverine phytoplankton endmember values of 30.5 ±2.5‰ in the Lena river. Also, we tested running the mixing model using a $\delta^{13}C$-POC value (-32.68 ±2.00 ‰, n = 9) as primary production end-member, which lowered the primary production contribution to the DOM flux budget by about 10% (from 21 % to 11 %). The outcome of endmember mixing models is limited by the quality and uncertainty of the end-member values, and we think we made the best possible calculation, with primary production likely lying between 11 % and 21 %.*

*We thank the reviewer for her/his critical note and based on our answer above, we will change or edit the following:*
- *Methods: incorporate values from literature (e.g. Winterfeld et al., 2015)*
- *Discussion: elaborate on the uncertainty and why we choose to use $\delta^{13}C$-DOC instead of $\delta^{13}C$-POC. This is because we are interested mainly in tracing the DOM component in lateral transport from soils to the outlet. We considered it incorrect to then use POC as an endmember because fractionation effects can lead to significant differences between the dissolved and particulate fractions and we also used dissolved tracer values for the soil endmembers.*
- *Discussion: Suggestion for follow-up research: extensive in stream sampling of algal mats and leachates of algal mats in whole water samples.*

**Specific Comments**

Is the definition for dissolved in this study 0.7 mm? This needs to be clarified in the Methods.

*We agree with the reviewer, we inserted at the beginning of paragraph 2.5: "All DOC samples have been filtered through glass fiber filters with 0.7 $\mu$m nominal pore size."*

Line 52: "With raising arctic temperatures"

Remove "with" for proper sentence structure, "raising" should be "rising"

*Thank you for noticing this mistake.*

Line 60: Remove "a" before 18%

*Removed typo.*

Line 88: The authors state that the middle and small catchments are exclusively underlain by permafrost, but then provide percentages of the catchments underlain by permafrost (60% and 73%). The word exclusively is incorrect and the sentence needs to be corrected.

*Our apologies, this should have been "almost exclusively", this will be corrected.*

Lines 115-116: which thermal layers and soil horizons, specifically? This information should be provided here.

*Edited to: "permafrost and active layer, organic and mineral horizons, and HCP, LCP and flat polygon types"*

Lines 118-120: There are many sentences throughout the introduction that need to be broken from a single run-on sentence to multiple sentences for clarity. This sentence in particular would benefit from splitting into: 1) The focus on LCP/HCP differences, then 2) LCP and HCP "imprints" on stream water OM composition and flux.

*Thank you for your comment on the clarity of sentences. Since the introduction needs a definite upgrade this comment's suggestion will be applied in rewriting the introduction.*

Line 122: Any ways to reduce verbiage will improve messaging and clarity: phrases like "circumpolar small coastal watersheds" take a fair amount of time to interpret. Does it matter that the watersheds are coastal? If so, how? Would small, polar watersheds suffice? You have already highlighted the spatial extent of small watersheds in the Arctic, so "circum" doesn't add much as a descriptor here.

*This is a good point as well, we will reduce the unnecessary usage of prepositions.*

Line 124: Again, while these types of watersheds cover much of the Arctic, you are only considering a single watershed. I suggest more caution in extrapolating results from a single watershed to the "circum-Arctic".

*We agree that going from a small (4 km$^2$) watershed to a region or even circum-Arctic extrapolation is bold. In our opinion this study area is highly representative of a system that is abundant along the arctic coastline (i.e. IWP tundra terrain), we will put more emphasis on the areal extent of this type of terrain within the circumarctic watershed (~30 %). Yet it is important for us that the point of studying this type of catchment comes across: The IWP tundra system is historically important as a carbon sink and under warming conditions which seems to be transitioning into a carbon source. To assess the lateral carbon flux component of this transition we must study this type of watershed in order to understand what is happening at a large scale. We acknowledge that we can make clear that we are studying the IWP catchment system in a local setting and remove mentions that hint toward panarctic implications.*

Line 146: I recommend adding: "precipitation means **are** 254 and 161 mm**, respectively**"

*Thank you. This has been added.*

Line 155: Remove comma between "Both, winter". Correct future instances with this same convention used.

*Thank you, this must have been a typo.*

Line 155: Have winter and summer sea ice extent **gradually** declined? Gradual does not seem the best choice to explain long-term sea ice dynamics in the region.

*Agreed, changed into 'rapidly'.*

2.8 DOM optical properties

This section needs to be expanded so readers have a better understanding of how samples were measured and processed. Optical properties are the basis for composition, so it's an important section. Were optical measurements made in duplicate, triplicate?

*Optical indices were measured on single samples for all but the incubation samples. The incubations were run in triplicate hence the optical properties were derived in accordance. We have added these details to the method section.*

Line 276: Turned regularly – about how often? Once per day?

*"Once a day", edited in text.*

2.9.1

So, for each stream, a total of 9 vials were incubated? (3 vials for each time period sampled?) This should be more clearly indicated. Was there a control?

*This we will clarify in the text: "From three tributary streams (A, B and C) we incubated whole water samples of 60 mL in triplicate for 0 (baseline) 7 and 14 days. This incubation was repeated at three different starting times (a dry and warm day, a cold and dry day, a cold and wet day) during the sampling period to account for natural variability within the streams. Hence a total of 81 vials were incubated and analyzed. In addition, we 'incubated' a distilled water sample as a control for the method used".*

2.9.2

Were these run in triplicate again? Sampled only for DOC, or also optics?

*Thank you. Indeed, porewater incubation samples were also run in triplicate (edited in text) but not analyzed for optics as there was not enough sample volume available.*

3.2.2 First paragraph (lines 372 -378)

Since the differences in OC between active layer and permafrost are statistically significant for LCP and HCP, I would group these together and separate the description of the flat group, where active layer and permafrost were not statistically different.

*Thank you, we implemented this improvement.*

Lines 393-394: The tense changes between these sentences; ensure that the same tense (either past or present) is used throughout. This happens in other instances in the manuscript.

*We will aim to correct inconsistency in the next version of the manuscript.*

Line 419: Swap locations of "from concentration" to read "concentration from"

*Corrected, thank you.*

Line 419-420: This sentence needs to be restructured, it is difficult to read.

*Changed into: "In contrast to that, DOC losses in the Oi-horizon were significantly higher (17.3 ±16 %, n = 3, T = 7 days)."*

Line 421: "begin" should be "beginning"

*Corrected.*

Lines 434-436: This sentence is contradictory. "Slope ratio (Sr), which negatively correlates with MW of DOM, was negatively correlated with SUVA254 in our porewater samples while

Sr shows a weak positive correlation with SUVA254 (i.e. lower MW molecules were less aromatic)." What are the authors referring to for each statement?

*We apologize for this mistake, the corrected sentence now is: "Slope ratio (Sr), which negatively correlates with MW of DOM, was negatively correlated with SUVA$_{254}$ in our porewater samples (i.e. lower MW molecules were less aromatic). Contrastingly Sr shows a weak positive correlation with SUVA$_{254}$ (i.e. lower MW molecules were more aromatic) in stream water samples."*

Lines 478-479: Permafrost contribution increased 3%, active layer was constant and PP decreased by 4% (check math to ensure all changes are appropriately represented)

*Thank you for your comment; this is a rounding error I now added one decimal precision but an error of 0.1 % remains.*

Line 521: "overlaying" should be "overlying"

*Corrected.*

Line 553: Fix "An explanation **is be** the presence"

*Corrected.*

Line 575: Remove "in" before "in situ"

*Corrected.*

Line 583-584: "A relatively small proportion of the total SOM pool enters the aquatic system as DOM"

Do you have a reference to back up this statement?

*Thank you for your comment. This statement is supported by the DOC yields presented in this paper (up to 10 mg DOC per gram soil C) and by other work (e.g. Wickland et al., 2018 ERL). However, I realize that this statement is not adding much in this part of the paper hence it will be removed.*

Line 614-619: I would be shocked if the short-term, large variability in CDOM was due to primary production. At the outlet, it seems much more likely that the coastal ocean/lagoon waters are driving variability in CDOM, which would also relate to temperature changes assuming there is a relatively large temperature gradient between stream and marine waters. This also speaks to potential issues with time series observations at the outlet. It seems there is a marine influence. How representative are these measurements of terrestrial inputs versus influenced by processes in the lagoon?

*Thank you for your comment. The only moment when the measurements were influenced by tidal fluctuations was during a storm surge on the 16$^{th}$ and 17$^{th}$ of August, when CDOM concentrations showed a sharp decline. This was mentioned in the text. We checked the automatic water sampler multiple times each day and concluded that with exception of the*

*storm surge there was no interaction with the lagoon (i.e. there was an elevation difference with the exact outlet of at least a meter where water was shooting/free flowing).*

Lines 640-656: The authors need more data to identify primary productivity within streams as such a significant driver of DOC dynamics. From what is presented, this appears a hypothesis at best. If the data is there to support it, it needs to be presented in much clearer terms.

*We indeed have not done targeted sampling for algal or primary production (but observed algal mats). Instead we have used compositional tools to perform source contribution calculations through endmember mixing. Three sources are identified and their presence in the stream is quantified with mixing analysis. We realize this is an indirect approach, but it is widely used. Nevertheless, we will include references that use similar methods and exemplify the presence of primary produced material, and also emphasize the need for targeted sampling in the future.*

Line 663: I don't think "hiatus" is the correct word choice here

*Here hiatus aims to exemplify the gap between modeled/measured permafrost contribution in small streams and large river. We agree that 'difference' might be a better word to use.*

Line 675-677: I don't disagree with these statements, but they highlight how little is known with a 10-day sampling timeframe. This is a much-needed dataset for the region, but the conclusions being drawn seem to be overextending the meaning in the data. The data is important on its own, without trying to extend to conclusions that can't be drawn with any significant confidence.

*We agree that extrapolation on the sole basis of the data presented here is not a substitute for larger scale, longer term, more in-depth research. Yet we present the 'rough upscaling' as a first estimate. We will make this clearer in the manuscript.*

Line 689-690: "Hence, our data suggest that a large fraction of (labile) DOM may be utilized before reaching the stream network."

This wasn't abundantly clear in the Results or data provided – where is this conclusion coming from?

*The conclusion follows from the relative contribution of terrestrial material in riverine DOM following the source apportionment in combination with the large difference in DOC concentration between the soils and the streams. Also, our incubations show soil BDOC between 5 % and 17 % after 7 days. Although this isn't high, we assume that transport times in this flat terrain are likely very long and thus significant degradation can take place. We will further clarify this in the manuscript.*

Line 693: "mostly weather and hydrology driven"

I agree with this, but it conflicts with earlier statements about the importance of in-stream primary productivity. It is hard to rectify how weather and hydrology control organic carbon dynamics, but portions of that cycling (i.e., CDOM) are controlled by other factors.

*We think these are two different things, and do not really see how this is conflicting. Overall, the mixing model suggests that primary production contributes about 21 % on average and between 14 and 31% of the total stream DOC pool over the course of the monitoring period. But the largest fluctuations in bulk CDOM, DOC and POC concentrations are caused by the storm events (Fig. 3) and that is why we here write that it is mostly weather and hydrology driven.*

Line 711: "which is probably an overestimation"

I don't think these numbers should be presented. While the data collected here is important, it covers a single system for 10 days in a single year. Any extrapolation of those results to a larger area is fraught with uncertainty. By providing this number, you are implying some confidence in it, and it is likely to be used in the future (e.g., for modeling) with the required uncertainty detached from the number.

*We value this comment, and will make it clearer that these numbers come with high uncertainty yet are a first best guess. Even though we see the reviewers' points that future studies may use this, we cannot be responsible for their usage of our data.*

Lines 713-715: Again, the data collected here doesn't really support these large, general claims. It is more suitable to focus on what you observed. Expanding the relevance needs to be very careful with caveats attached.

*We will reconsider the framing of the statements and alter the manuscript in such a way that it becomes clearer that we are only making claims based on our observations in combination with what is already published.*

Lines 733-735: I don't think the reference to negative priming is relevant here.

*We agree and will remove the reference and sentences about negative priming.*

Line 742: "Moreover, there is a simple need to map these watersheds at the basis of the aforementioned challenge."

This sentence is unclear.

*Changed to: "Moreover, there is a need for more detailed mapping of these watersheds (e.g. high-resolution watershed delineation and assessment of watershed characteristics)."*

Line 748: "These techniques together with standardization of methods are there for recommended for a harmonized approach on understanding lateral permafrost-OM-dynamics."

This sentence is unclear.

*Changed to: "Usage of these techniques together with standardization of methods are thus recommended for a harmonized approach toward understanding lateral permafrost-OM-dynamics."*

Line 768: I wouldn't define a 10-day sampling window as "long-term"

*Here we mean continuous monitoring in the general sense, it can be 10 days like we did, but ideally even longer. Have changed it into: "achieved by long-term (weeks or ideally months) monitoring ..."*

Line 780-782: There is no discussion of remote sensing throughout the paper, only a single mention of it. And really, remote sensing tools aren't suitable for observing stream OM dynamics – the spatial scale is too fine. I recommend removing this sentence.

*Thank you for this comment. We mention remote sensing with the aim at using it for the detection of landscape scale changes (e.g. polygon degradation on a large scale) because we think this could help mapping regions where significant change is underway. Remote sensing techniques are also being used in predicting SOC stocks hence why not for soil DOC stocks. Knowing the areas of change together with predicted stock can help predict stream DOC. Or at least help better understanding the controls on stream DOC flux for instance. We will make this clearer and mention the example of remote sensing between brackets in the manuscript.*

Figure 1

The contrast in the upper panel needs to be improved, and it appears that the inlay indicated in the top panel (black box) does not correspond to the area shown in the lower panel.

*Will be altered. The box has somehow moved place. Our apologies.*

Figure 5

This figure would be improved by showing change in dC13 to correspond to changes in DOC shown on the left

*We will add another panel to show this. The current right panel is there to simply show the initial (T=0) d13C-DOC values and its variability in the watershed.*

Table 1

While the summary is appreciated, I recommend removing this table based on the length of the paper and presence of this information in Methods and references therein.

*We decided to keep this as is.*

---

## Referee Report (RR1)

Review Speetjens et al Biogeoscience

**Resubmission - Dissolved organic matter characterization in soils and streams in a small coastal low-arctic catchment**

**General Comments:**

In this manuscript by Speetjens et al they have addressed all comments from the previous round of revision and clarity has improved. The paper continues to be a real asset in C Biogeochemistry especially in Arctic and small coastal watersheds. However, there are some minor improvements that could be made in the introduction to polish a bit more the central idea of the paper.

The introduction has improved and now the importance or relevance of these small watersheds is clearer. But the central aim of the paper is to better understand terrestrial aquatic linkages and ocean OM budgets yet the context of these topics are not very clear in the Introduction leading up to the last paragraph where the authors lay out the objective of the paper. This suggestion is not intended to be a major overhaul, just add a few more sentences that help to contextualize lateral fluxes and OM budgets, which are already large black box in terms of OM processes regardless of geographic location. In the Arctic these fluxes have a much greater importance in terms of C cycling and delivery to the ocean but are also a lot more vulnerable to change (whether it is to be enhanced or inhibited due to thaw). Perhaps this sort of rationale is missing. For example, something along the lines of the first sentences in sections 4.4 would be great to include in the introduction prior to the objectives paragraph.

There some citations that are not in the References section, they were probably removed during the editing process. For example, Couture et al., 2018; Couture & Pollard, 2017, Dunton et al., 2006 are missing.

Lines 160 – The phrase "Under current, warming, climate conditions" perhaps there are too many commas?

Line 365 – 47mm is mentioned twice, maybe just needs to be mentioned once?

---

## Author Response (AR2)

**Response to reviews:** "Dissolved organic matter characterization in soils and streams in a small coastal low-arctic catchment" by Speetjens et al.

**Comments to the author**:

Reviewer 1:
The authors have done a very nice job editing the paper. The Introduction and Discussion are quite good, in line with the high-quality datasets collected. I have a few minor, technical edits suggested, and one more important change that I recommend as follows.

*Thank you very much for your positive response; we have revised the manuscript as follows, answers to comments are below each comment in italic font.*

Lines 665-672: Right now, you are accounting for storms in your mean value, and then also commenting on the additional POC flux due to storms. I think it would make more sense to provide a baseline POC flux, with storms excluded from the mean values, and then highlight the impact of the storms. Right now, the numbers you present are effectively counting the impact of storms on POC flux twice.

*Thank you for this insight. The means, standard deviations and flux calculations have been calculated anew.*

This section as a whole (4.4) is quite good now, I agree with the author's decision to keep this estimate in place.

*Thank you.*

Specific Suggestions
Line 595: "drawdown" should be "dilution", drawdown implies degradation or consumption

*We implemented this suggestion.*

Line 658: "Arctic ocean" - ocean should be capitalized

*Corrected*

Line 665: I suggest "area based estimated OC flux for the region." is reworded to "area-based OC flux estimate for the region."

*Corrected*

Line 755: I suggest "more geographically widespread and longer timespan covering monitoring efforts" is reworded to "more spatially and temporally widespread monitoring efforts"

*Corrected*

Line 756: I suggest "are implemented, e.g. through sensor installations, use of cost-effective optical proxies to monitor change" is slightly altered to "are implemented (e.g., through sensor installations, use of cost-effective optical proxies) to monitor change. This would help

emphasize the last 3 words are part of the original sentence, and separate from the examples given.

*Thank you for this helpful comment, we implemented the improvement.*

I recommend the authors include the description of the difference in height at the outlet limiting impact of exchange with the lagoon at some point in the manuscript (there are a few suitable locations to include, preferably sooner than later in the text). Something similar to what was included in the author response would suffice - "there was an elevation difference with the exact outlet of at least a meter where water was shooting/free flowing". Admittedly, while I have been in these environments, I was envisioning a more direct connection with the lagoon so this is an important detail to include for clarity.

*This is a good point, we inserted the clarification in section 2.3, where we introduce the sampling at the outlet.*

Reviewer 2:
General Comments:
In this manuscript by Speetjens et al they have addressed all comments from the previous round of revision and clarity has improved. The paper continues to be a real asset in C Biogeochemistry especially in Arctic and small coastal watersheds. However, there are some minor improvements that could be made in the introduction to polish a bit more the central idea of the paper.
*Thank you very much for recognizing the value of our work and for sharing your improvements with us. Responses to suggestions are included in italics below each comment:*

The introduction has improved and now the importance or relevance of these small watersheds is clearer. But the central aim of the paper is to better understand terrestrial aquatic linkages and ocean OM budgets yet the context of these topics are not very clear in the Introduction leading up to the last paragraph where the authors lay out the objective of the paper. This suggestion is not intended to be a major overhaul, just add a few more sentences that help to contextualize lateral fluxes and OM budgets, which are already large black box in terms of OM processes regardless of geographic location. In the Arctic these fluxes have a much greater importance in terms of C cycling and delivery to the ocean but are also a lot more vulnerable to change (whether it is to be enhanced or inhibited due to thaw). Perhaps this sort of rationale is missing. For example, something along the lines of the first sentences in sections 4.4 would be great to include in the introduction prior to the objectives paragraph.

*This is a good suggestion, we have added the following to the introduction: "Lateral OM fluxes through and from inland waters are still poorly constrained (Drake et al., 2018), particularly the export from smaller basins draining directly into the ocean. These small basins are at the same time very relevant for OC cycling as their soils are rich in carbon stocks, and also particularly vulnerable to current climate warming that triggers permafrost thaw leading to changes in hydrology and biogeochemistry. Due their abundance and proximity to the Arctic Ocean, IWP tundra streams have the potential to export large quantities of terrestrial OM into coastal waters." …. "and the effect of thawing permafrost herein (e.g. enhancing or inhibiting)." – from line 99 onward.*

There some citations that are not in the References section, they were probably removed

during the editing process. For example, Couture et al., 2018; Couture & Pollard, 2017, Dunton et al., 2006 are missing.

*This was corrected, thank you.*

Lines 160 – The phrase "Under current, warming, climate conditions" perhaps there are too many commas?
*We removed the commas.*

Line 365 – 47mm is mentioned twice, maybe just needs to be mentioned once?

*We removed the duplicate mention.*

---

## Author Response (AR3)

Response to editor:

The editor suggested to remove paragraph breaks in the abstract.

*We implemented this change.*